# Mitochondrial impairment and intracellular reactive oxygen species alter primary cilia morphology

Noah Moruzzi[1,2,3] (iD), Ismael Valladolid-Acebes[1], Sukanya A Kannabiran[2,3,4], Sara Bulgaro[1], Ingo Burtscher[2,3], Barbara Leibiger[1], Ingo B Leibiger[1], Per-Olof Berggren[1], Kerstin Brismar[1]

Primary cilia have recently emerged as cellular signaling organelles. Their homeostasis and function require a high amount of energy. However, how energy depletion and mitochondria impairment affect cilia have barely been addressed. We first studied the spatial relationship between a mitochondria subset in proximity to the cilium in vitro, finding similar mitochondrial activity measured as mitochondrial membrane potential compared with the cellular network. Next, using common primary cilia cell models and inhibitors of mitochondrial energy production, we found alterations in cilia number and/or length due to energy depletion and mitochondrial reactive oxygen species (ROS) overproduction. Finally, by using a mouse model of type 2 diabetes mellitus, we provided in vivo evidence that cilia morphology is impaired in diabetic nephropathy, which is characterized by ROS overproduction and impaired mitochondrial metabolism. In conclusion, we showed that energy imbalance and mitochondrial ROS affect cilia morphology and number, indicating that conditions characterized by mitochondria and radicals imbalances might lead to ciliary impairment.

## Introduction

The primary cilium is a protrusion from the cellular membrane that can act as sensor for the extracellular environment. These non-motile cilia are involved in several signaling pathways during development and tissue homeostasis (Wheway et al, 2018). In invertebrates such as *Drosophila melanogaster* and *Caenorhabditis elegans* primary cilia are mainly found in sensory neurons, whereas in higher vertebrates roughly 80% of the cells are ciliated. The primary cilium has a typical structure with a ring of nine microtubule pairs in the outer part of its cross section (so called structure "9 + 0"), whose assembly and retraction begins from their ciliary tip (Satir et al, 2010). Transport mechanisms into the ciliary compartment are essential for cilia formation and maintenance by carrying proteins to and from the basal body, even though protein translation directly within motile cilia has been recently shown (Hao et al, 2021). In both cases, the protein cargoes are actively transported along the axoneme by multimeric protein complexes called intraflagellar transport proteins (IFTs). IFT-B mediates anterograde transport to the ciliary tip by binding cargoes to specific kinesin-II motor proteins, whereas retrograde transport to the basal body is dependent on IFT-A and cytoplasmic dynein I.

Kinesin (anterograde) and dynein (retrograde) motors hydrolyze ATP for their movement, whereas proteins responsible for sorting and trafficking located at the basal body use GTP (Yoshimura et al, 2007; Fan & Margolis, 2011). For these reasons, cilium maintenance is energetically costly. However, the relationship between primary cilia and energy scarcity, especially regarding mitochondrial metabolism, is unclear. The mitochondrion is responsible for up to 80–90% of cellular ATP production (Szewczyk & Wojtczak, 2002; Gilkerson et al, 2003), is a center for several catabolic and anabolic processes and has emerged as a signaling organelle (i.e., production of reactive oxygen species [ROS]) (Turrens, 2003; Brand, 2016). Thus, its impairment can affect in multiple ways ciliary assembly and functions. A recent study, addressing the possibility that defective mitochondrial genes can potentially lead to ciliary impairment and heterotaxy, found that increasing or blocking mitochondria energy production caused deleterious consequences during development (Burkhalter et al, 2019). Surprisingly, decreasing ATP content by manipulation of different genes as well as Complex I (CI) inhibitor rotenone, increased cilia length at the embryonic stage and in human fibroblasts, raising the possibility that a decrease in energy level can elongate the cilium (Burkhalter et al, 2019). An increase in cilia length has also been shown after a single dose of mitochondrial inhibitor MPTP in dopaminergic neurons in mice (Bae et al, 2019).

In this work, we aim to clarify the relationship between primary cilia and mitochondria, with special focus on how mitochondrial function, its byproducts and cellular energy balance affect ciliary morphology.

[1]The Rolf Luft Research Center for Diabetes and Endocrinology, Karolinska Institutet, Karolinska University Hospital, Stockholm, Sweden [2]Institute of Diabetes and Regeneration Research, Helmholtz Zentrum München, Neuherberg, Germany [3]German Center for Diabetes Research (DZD), Neuherberg, Germany [4]Department of Biochemistry and Molecular Cell Biology, University Medical Center Hamburg-Eppendorf, Hamburg, Germany

Correspondence: noah.moruzzi@ki.se

# Results and Discussion

### Mitochondria are in close relationship with the basal body

Primary cilia formation and maintenance have been long thought to use a high amount of cellular ATP, which is mainly produced by mitochondria. Similar to primary cilia, mitochondria have a typical diameter of about 200 nm and cannot fit inside the ciliary structure. Thus, mitochondria must be located in close proximity to the cilium where they provide ATP, which can be equilibrated in the cilium using adenylate and creatine kinase shuttle (Acevedo et al, 2019). More than 1,000 proteins are classified in the ciliary proteome, which combines results from high throughput proteomics, differential expression and comparative genomics studies (CiliaDB [Arnaiz et al, 2009]). Searching the database of predicted or bona fide ciliary/basal body proteins, we found hits such as subunits of mitochondrial complexes as well as transporters involved in ATP generation and electron transport chain from a variety of organisms including humans (Table 1).

To investigate the spatial relationship between the primary cilium and the mitochondrial network, we used *mouse inner medullary collecting duct cells* (IMCD3), a widely used cell line in cilia research. To be able to visualize cilia and mitochondria contemporary in live cell, we generated stable IMCD3 cell lines over-expressing a cilia-specific protein, ADP-ribosylation factor-like protein 13b (ARL13b) tagged with fluorescent proteins Venus or red fluorescent protein (RFP). Whereas in the ARL13b-RFP we expressed a photoactivatable GFP targeted to the mitochondrial matrix (Mito-PAGFP) (Fig 1A), in the ARL13b-Venus IMCD3 line the mitochondria were visualized with the potentiometric dye tetramethylrhodamine ethyl ester (TMRE) or Mitotracker DeepRed (Figs 1B and S1A, respectively). In IMCD3 cells, the primary cilia were distributed in the apical area of the cell and the mitochondria in a network around and above the nucleus. However, we observed a distinct subset of mitochondria in proximity to the cilium that was disconnected from most of the cell mitochondrial network (Figs 1A and B and S1A). In line with our finding, Burkhalter et al (2019) recently provided some electron micrographs showing the possible tubular connection between some mitochondria and the base of the primary cilium (Burkhalter et al, 2019). To test if the subsets of mitochondria close to the base of the cilium displayed different bioenergetic properties compared with the rest of the mitochondrial network, we measured their transmembrane potential (Ψm) using TMRE in ARL13b-Venus IMCD3 cells (Fig 1B). Because within a distance of 2 $\mu m$ away from the ciliary base (identified with ARL13b) we observed most of these mitochondria disjointed from the perinuclear network, we set this distance as arbitrary threshold to compare the Ψm of the mitochondrial subset in proximity to the cilium to the average Ψm of all cellular mitochondria. Here, we did not detect any significant changes (Figs 1C and S1C), suggesting that the mitochondrial bioenergetics in the network proximal to the cilium was similar compared with the one of the rest of the cell, indicating that this subset is in equilibrium with the intracellular mitochondrial network.

### Blocking complex I (CI) or ATPase decrease energy levels and ciliation in IMCD3

Primary cilia assembly and protein transport to and along the axoneme has been described and partially characterized (Broekhuis et al, 2013). For its maintenance, it has been speculated that ciliary machinery requires a high amount of energy. However, to what extent energy scarcity and impairment of the main producer of cellular ATP (i.e., mitochondria) affect ciliary assembly and morphology, potentially altering ciliary functions, is not clear. A recent study suggested that decrease in ATP levels might lead to ciliary elongation and defective signaling in human fibroblasts and

**Table 1. List of the 10 most represented mitochondrial proteins found in the ciliary database ordered by ciliary evidence (CiliaDB [Arnaiz et al, 2009]).**

| Name | Ciliary evidence | Organisms with ciliary evidence | Non flagellary evidence | Proteomic studies | Evidence in *H. sapiens* |
|---|---|---|---|---|---|
| Ubiquinol-cytochrome c reductase core protein II | 6 | 4 | 3 | 3 | 0 |
| NADH dehydrogenase (ubiquinone) Fe-S protein 1, 75 kD (NADH-coenzyme Q reductase) | 5 | 5 | 2 | 1 | 1 |
| ATP synthase, H+ transporting, mitochondrial F1 complex, beta polypeptide | 5 | 5 | 1 | 1 | 1 |
| Creatine kinase, mitochondrial 1B | 5 | 3 | 2 | 2 | 1 |
| Isocitrate dehydrogenase 2 (NADP+), mitochondrial | 5 | 5 | 3 | 3 | 1 |
| Inner membrane protein, mitochondrial | 5 | 4 | 3 | 3 | 1 |
| Acyl-CoA dehydrogenase, short/branched chain | 5 | 2 | 3 | 0 | 1 |
| Voltage-dependent anion channel 2 | 5 | 4 | 4 | 3 | 0 |
| Solute carrier family 25 (mitochondrial carrier; phosphate carrier), member 3 | 4 | 4 | 4 | 2 | 1 |
| Solute carrier family 25 (mitochondrial carrier; dicarboxylate transporter), member 10 | 4 | 3 | 3 | 3 | 0 |

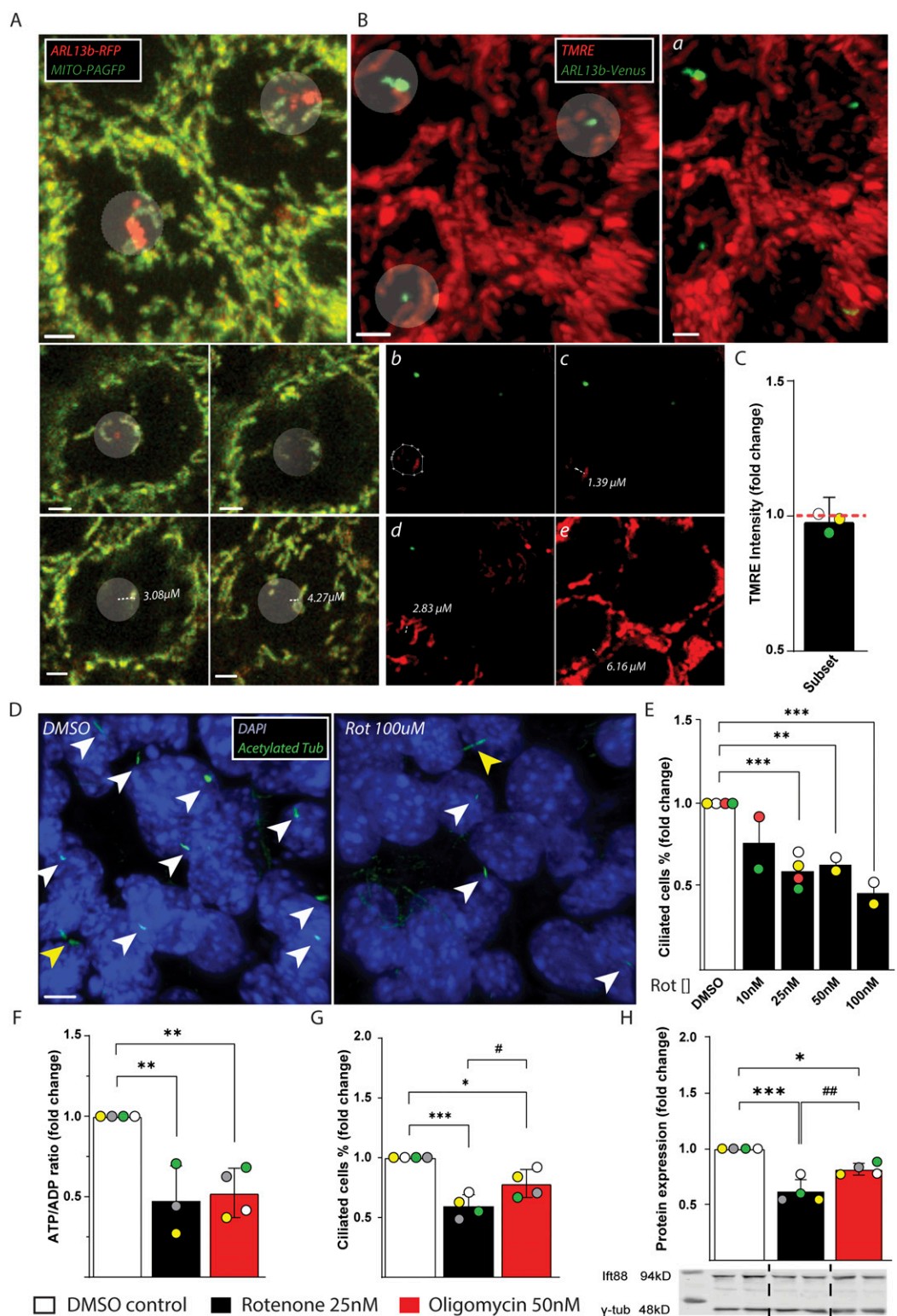

**Figure 1. Mitochondria relationship with the cilia and effect of mitochondria inhibitors on IMCD3 ciliation.**
**(A)** Representative image of live recording of stable IMCD3 ARL13b-RFP-Mito-PAGFP at confluence. The transparent white circles indicate the radius of 2 $\mu$M in the xy dimensions from the base of the cilia. The below images represent slices of the z dimension to show the mitochondria location within the cell. The dashed lines show the distance between the start site of ARL13b and the location of that particular mitochondria indicated with the +. The measurements where performed with Imaris which allows to draw lines in between z-stack and measurements of the distance between two different points in 3D which are reported in the figures. Scale bar = 2 $\mu$M. **(B)** Representative image of live recording of stable IMCD3 ARL13b-Venus loaded with 25 nM TMRE. The transparent white circle/ellipse indicates the mitochondria within the radius of 2 $\mu$M from the base of

zebrafish embryos (Burkhalter et al, 2019). Vice versa, the reduction in cellular energy could decrease the activities of motor proteins that use ATP such as kinesin, dynein, and GTPases involved in transport and import of ciliary proteins (Yoshimura et al, 2007; Fan & Margolis, 2011), thus decreasing ciliary assembly.

To clarify the association between mitochondrial function and ciliary homeostasis we first treated IMCD3 cells with increasing concentrations of the inhibitor of mitochondrial complex I (CI) rotenone. After 48 h of incubation with rotenone the number of ciliated cells decreased in a dose-dependent manner (Figs 1D and E and S1D). To test whether the decrease in ciliation was related to other rotenone off-target effects, we measured cilia number and cell energy status after 48 h incubation with rotenone (25 nM) and in a parallel treatment with the inhibitor of ATPase oligomycin (50 nM), which decreased the cellular energy status (ATP/ADP ratio) to a similar extent (Figs 1F and S1E). According to the view of direct relationship between energy status and cilia assembly, not only rotenone, but also oligomycin treatment was able to reduce the amount of ciliated cells (Figs 1G and S1F). This was reflected by a similar decrease in cellular levels of intraflagellar transport 88 (IFT88) protein compared with control (Figs 1H and S1G). However, ciliated cells are decrease in lesser extent by oligomycin treatment compared with rotenone (Figs 1G and S1F). Our results suggest that decreasing mitochondria ATP production in IMCD3 cells impairs cilia morphology. However, decreasing equally the mitochondria energy production by either blocking CI or ATPase exerted differential effects. In fact, CI inhibitor rotenone-induced a stronger reduction of ciliated cells compared with the ATPase inhibitor oligomycin, suggesting an additive mechanism to impair cilia morphology other than reducing cellular energy status.

### ROS are partly responsible for decreased number of ciliated IMCD3 cells

Rotenone binding to the CI causes overproduction of superoxide anion thus increasing intramitochondrial ROS (Brand, 2016). However, depending on the metabolic state of the cell, also oligomycin can induce ROS formation (Turrens, 2003). Thus, the lower amount of ciliated cells using CI inhibitor rotenone led us to investigate ROS production as potential additive effect impairing the primary cilium. Thus, we measured mitochondrial superoxide and cytosolic ROS using MitoSOX and dichlorofluorescin diacetate (DCFDA), respectively. In IMCD3, we found that only rotenone increased the intramitochondrial radicals (Figs 2A and S2A and B), but

both inhibitors increased the cytosolic radicals to a similar extent (Figs 2B and S2C and D). This is probably due to the reduced state of the electron transport chain, which can lead to ROS production in the outer part of the mitochondrial membrane (from CIII), whereas the ROS produced by rotenone treatment are released from CI into the mitochondrial matrix (Turrens, 2003).

To understand if the cytosolic ROS are responsible for impaired ciliation, we treated the cells with an external oxidative stress-inducing agent, namely tert-butyl hydroperoxide (TBOOH). After 48 h, the number of ciliated cells was unaffected using concentrations up to the point of which the compound induced cell death (100 $\mu$M) (Fig S2E). However, any of the TBOOH concentrations were able to reproduce the redox imbalance that we found using mitochondrial inhibitors (Fig S2F). Using a different radical inducer such as paraquat, we obtained similar results as with TBOOH (Fig S2G). As evidence suggested that ROS are involved in decreased cell ciliation we then opted for the opposite approach by quenching the radicals induced by rotenone with a combination of hydrosoluble (ascorbic acid and Tiron) or a liposoluble (retinoic acid) antioxidants. By administration of these compounds in combination with rotenone for 48 h, we found a significant maintenance of ciliated cells with the hydrosoluble antioxidant mixture and as tendency with retinoic acid, compared with rotenone alone (Figs 2C and S2H). The fact that this approach was not able to completely restore the cilia at the level of the controls, suggests that both cellular energy depletion and ROS are responsible for ciliary impairments. We investigated other possible events that could trigger the reabsorption of the cilium such as cell cycle and apoptosis (Kasahara & Inagaki, 2021). Here, we found a slight significant decrease in G1 phase and increase in the G2M phase after rotenone treatment (Figs 2D and S2I). Rotenone and oligomycin treatment did not increase cells apoptosis at the concentration we used, compared with control (Figs 2E and S2J).

In summary, although we found fewer IMCD3 cells in $G_1$ phase after rotenone and oligomycin treatments, the decrease of cell ciliation was mainly due to the intracellular ROS production and energy depletion. Accordingly, the effect of acute radical inducer treatment has been shown to decrease ciliation in MDCK canine kidney cells and after ischemic injury in kidney (Kim et al, 2013). Moreover, oxidative stress and reduced amount of antioxidant enzymes have been shown during atypical ciliopathy such as polycystic kidney disease (Maser et al, 2002) and depletion of peroxiredoxin 5 in IMCD3 is able to decrease ciliation (Ji et al, 2019) and cilia length (Agborbesong et al, 2022). Thus, kidney cells decrease ciliation or cilia length during conditions of reduced energy supply or oxidative stress.

---

the cilia. **(A, B, C, D, E)** The image plane was rotated using Imaris to show the mitochondrial subset close to the cilium. The dashed lines show the distance between the start site of ARL13b and the location of that particular mitochondria indicated with the +. Scale bar = 3 $\mu$M. **(C)** Mitochondrial membrane potential of mitochondria within <2 $\mu$m from the cilium compared with the overall cell mitochondrial network measured with TMRE in IMCD3 ARL13b-Venus stable cell line. Data are represented by mean + CI; n = 3. **(D)** Representative image of IMCD3 cell line treated for 48 h with DMSO vehicle or 100 nM of rotenone. Cilia stained with anti-acetylated $\alpha$-tubulin are indicated with white arrows, whereas not counted cytokinetic bridges in dividing cells are indicated with yellow arrows. **(E)** Quantification of dose-dependent effect of rotenone on number of ciliated IMCD3 cells. Data are represented by mean + SD; n ≥ 2. **(F)** ATP/ADP ratio in IMCD3 cells treated 48 h with DMSO vehicle, 25 nM rotenone or 50 nM oligomycin. Data are represented by mean + SD; n ≥ 3. **(G)** Quantification of IMCD3 ciliated cells treated as above. Data are represented by mean + SD; n = 4. **(H)** Representative WB image of IFT88 protein in IMCD3 cells treated as above. Quantification is represented by mean + SD; n = 4. Legend: Black empty = DMSO control; Black = rotenone 25 nM; Red = oligomycin 50 nM. Statistical significance between control versus treatments: $P < 0.05$ (*), <0.01 (**) or <0.001 (***). Rotenone versus oligomycin: $P < 0.05$ (#).

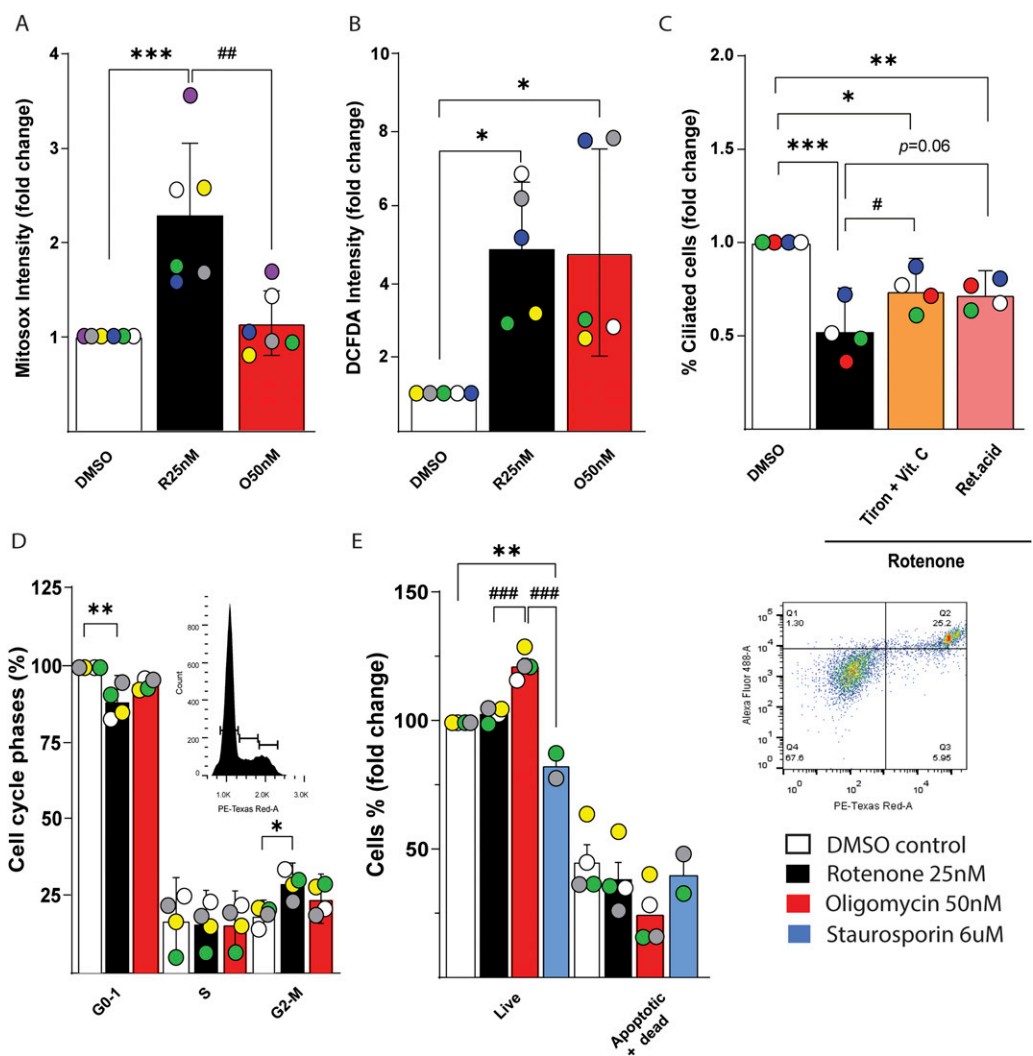

**Figure 2.  Reactive oxygen species, but not cells cycle and apoptosis, are in part responsible for decreased ciliation in IMCD3.**
**(A)** Quantification of Mitosox intensity in IMCD3 cells treated for 48 h with DMSO vehicle, 25 nM rotenone, or 50 nM oligomycin. Data are represented by mean + SD; n = 6.
**(B)** Quantification of DCFDA intensity in IMCD3 cells treated as above. Data are represented by mean + SD; n = 5. **(C)** Quantification of cilia number in IMCD3 cells treated for 48 h with 25 nM of rotenone in the presence of antioxidants (Tiron + Vit.C 1 mM, retinoic acid 1 $\mu$M). Data are represented by mean + CI; n = 4. **(D)** Cell cycle analysis in IMCD3 cells treated as above. In the inset is displayed a representative image of cell cycle plot in control cells. Data are represented by mean + CI; n = 4. **(E)** Analysis of apoptotic and dead IMCD3 cells treated as above normalized per 100 events of DMSO treated cells. In the inset is displayed a representative plot of control cells. Staurosporin 6 $\mu$M has been added 2 h prior the trypsinization as positive control. Data are represented by mean + SEM; n ≥ 2. Legend: Black empty = DMSO control; black = rotenone 25 nM; red = oligomycin 50 nM; blue = staurosporine 6 $\mu$M; orange = Tiron + Vit.C 1 mM; pink = Retinoic acid 1 $\mu$M. Statistical significance between control versus treatments: $P < 0.05$ (*), <0.01 (**), or <0.001 (***). Rotenone versus oligomycin: $P < 0.05$ (#), <0.01 (##), or <0.001 (###).

## Ciliation is impaired in kidneys of a diabetic nephropathy mouse model

To translate our findings of decrease in ciliation in IMCD3 cells in vitro, to the corresponding type of cell (inner medullary collecting duct cell) in the living organism, we took advantage of a diabetic mouse model namely Leptin-receptor–deficient (db/db) mouse, which develop diabetic nephropathy (Fig 3) (Cohen et al, 1995; Myakala et al, 2021). During diabetic nephropathy, increased oxidative stress and mitochondrial dysfunction have been described in the kidneys of db/db mice and humans (Brezniceanu et al, 2008; Forbes et al, 2008; Persson et al, 2012; Sharma et al, 2013; Sharma, 2016). At 8 wk of age the db/db mice are already obese,

hyperleptinemic, hyperglycemic, hyperinsulinemic compared with littermate controls (Fig S3A–H). However, whereas at 8 wk of age the kidneys of db/db mice display similar functions and morphology compared with lean controls, at week 24 the plasma creatinine and the kidney morphology indicate the progression of diabetic nephropathy (Fig 3A–C). At these two time points, we investigated the cilia number and morphology in two different cell types, that is, collecting duct and distal tubule cells. In the medullary collecting duct cells (Fig 3D) we found similar cilia morphology and number at 8 wk of age compared with wild-type littermates. However, at 24 wk of age, we found a significant decrease in cilia number (Fig 3E) and length (Fig 3F) in db/db mice. In line with our hypothesis that mitochondrial defects and/or excessive ROS alters cilia in IMCD, we

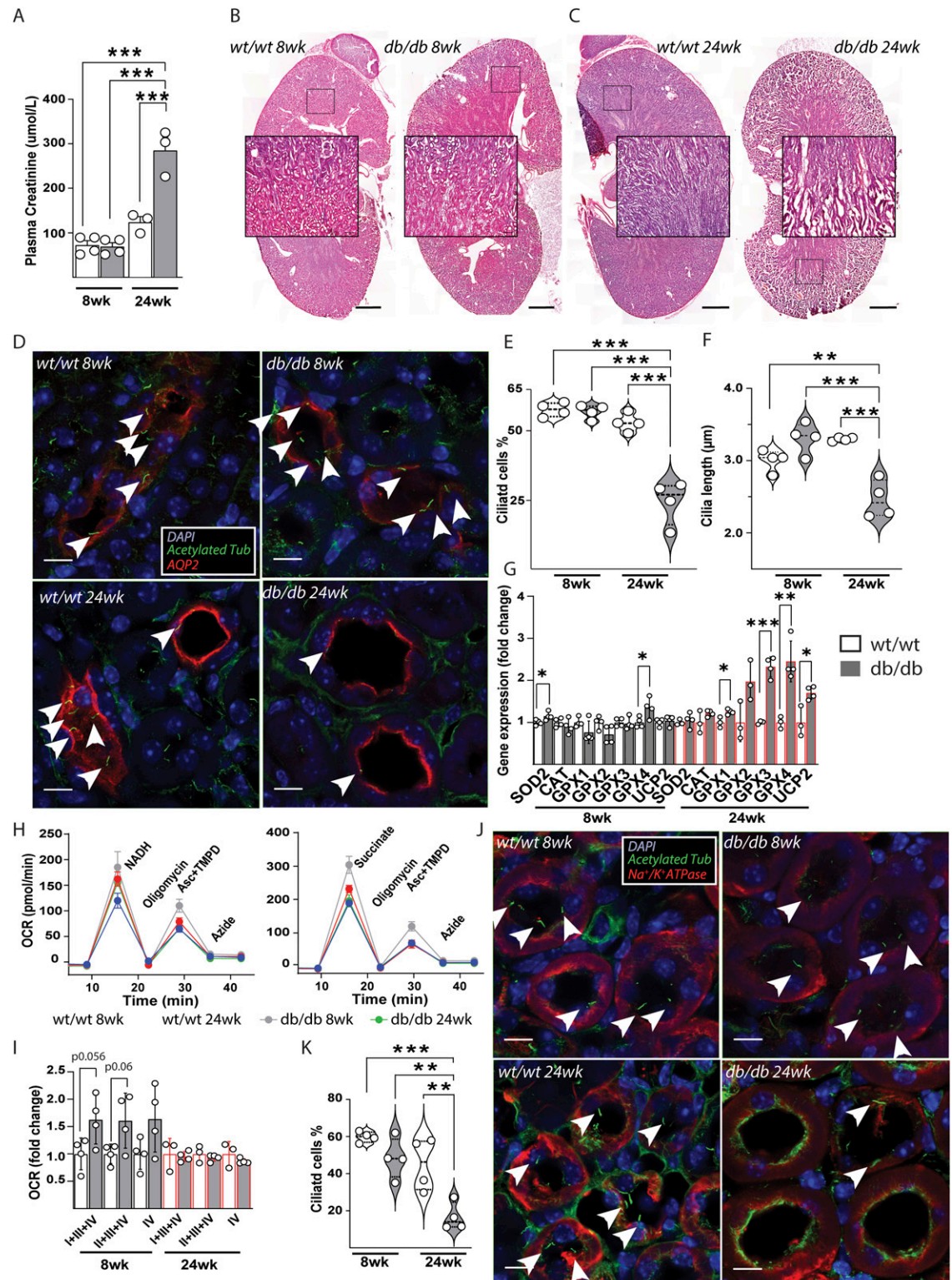

**Figure 3. Ciliation is impaired in kidney of diabetic nephropathy mouse model db/db.**
**(A)** Non-fasting plasma creatinine in wild-type and db/db mice at 8 (n = 4) and 24 (n = 3) wk of age. Data are represented by mean + SEM. **(B, C)** Representative morphological kidney sections (20 μm) counterstained with hematoxylin–eosin of 8- and 24-wk wild-type and db/db mouse. Scale bar = 1 mm in whole kidney pictures and 100 μm in blow up pictures. **(D)** Representative single stack images of kidney sections stained with ciliary acetylated tubulin (green) and collecting duct cells aquaporin-2 (red). Nuclei (blue) were counterstained with DAPI. **(E)** Percentage of ciliated collecting duct cells in wild-type and db/db mice at 8 and 24 wk of age. Data represent three medullary images (upper, middle, and lower kidney) in duplicates for each mouse (n = 4). Minimum 10 collecting duct cells have been counted for each

found that only 24 wk of age, the markers of oxidative stress, such as genes encoding for antioxidant enzymes and UCP2, were up-regulated in db/db mice (Fig 3G). In the same samples used to measure cilia morphology at 8 wk, we measured also the oxygen consumption from frozen samples to identify possible mitochondrial alteration (Fig 3H and I). Interestingly, we found a tendency in increased complexes I+II+IV, I+III+IV, and IV oxygen consumption at 8 wk of age when the cilia of db/db mice are similar to wild-type controls. This might be linked to the increase glomerular filtration rate (Gartner, 1978) and high amount of circulating substrates (lipids and glucose) in db/db (Bhargava & Schnellmann, 2017). This difference was not present at 24 wk of age where we found a similar oxygen consumption (Fig 3H and I). Whereas we cannot point out dysfunction of mitochondria or in energy production, these data suggests a decrease in mitochondrial activity between 8 and 24 wk in db/db mice. The increase in oxidative stress markers and in particular UCP2, encoding for a proton transporter which lowers mitochondrial ROS (Doke & Susztak, 2022), might be linked to mitochondrial as a source of the ROS which can affect cilia morphology in collecting ducts in db/db mice at 24 wk of age.

To understand whether the defective cilia morphology and number found in collecting duct was extended to other cell type in the kidney, we analyzed in the same manner cilia of distal tubules (Figs 3J and K and S3I). Tubular cells have been shown to have one of the highest mitochondrial densities, which is needed to generate a great amount of ATP for the active transport of molecules. In the kidney, and in particular in tubule cells, ROS accumulation has been shown to contribute to chronic kidney disease (Doke & Susztak, 2022). Here, we found a decrease in number of ciliated cells (Fig 3K) in 24-wk-old db/db mice without alteration of length in the cilia that were still present (Fig S2K), suggesting that cilia impairment during kidney nephropathy in db/db mice is not restricted to collecting duct cells.

Although a defective cilia morphology has been found in neurons during hyperleptinemia (Han et al, 2014; Kang et al, 2015), cilia were not impaired in the kidney of 8-wk-old db/db mice in presence of high leptin levels similar to 24-wk-old db/db mice. For the same reason, we excluded also other possible causes of ciliary derangements such as adiponectin, insulin levels, and altered lipid profile (Fig S3B–H). These results show that cilia morphology is altered concomitantly to diabetic nephropathy in at least two different cell types, and these changes are independent from levels of circulating hormones, that is, leptin and insulin and occurs at least contemporary to the increase of oxidative stress markers. A similar decrease in number of cilia was described for cells in pancreatic islets from diabetic models (Gerdes et al, 2014), which are also characterized by an increased mitochondria-driven oxidative stress as well as other cellular dysfunction (Eguchi et al,

2021). We cannot precisely discern whether mitochondrial defect/ ROS precedes ciliary impairments and longitudinal time course experiments on cilia and cellular phenotyping would be necessary to establish the exact timing of these events.

## Mitochondrial CI blockade elongates cilia in human retinal epithelial cells independently of cellular energy status

Distinct cell types might respond differently to energy depletion or oxidative stress depending on their ability to sustain energy production from glycolysis and detoxify radicals in a more efficient manner. To study a cell type different in these aspects to IMCD3 cells, we have analyzed another cell line known in ciliary studies namely *human telomerase reverse transcriptase immortalized retinal pigmented epithelial 1* (RPE1). RPE1 cells depleted of the hydrogen peroxide scavenger peroxiredoxine 5 have shown a reduction in cilia number (Ji et al, 2019). Surprisingly, using rotenone, we found a dose-dependent increase in maximum and median cilia length, whereas the minimum length was not affected (Figs 4A and B, S4A and B, and S5). In line with our findings but opposite to what we showed for IMCD3 cells, an increase in cilia length after mitochondria impairment with rotenone was shown during embryonic development and in human fibroblasts, which was suggested to be caused by a decreased reabsorption of the cilium because of ATP depletion (Burkhalter et al, 2019). To clarify if the primary mechanism responsible for cilia elongation in RPE1 was energy deprivation or oxidative stress, we first measured the effects of a dose-dependent blockade of CI and ATPase to define the rotenone and oligomycin doses that maximally inhibit the ATP linked respiration (Figs 4C and S4C–E). By measuring the oxygen consumption rate (OCR) and extracellular acidification rate (ECAR) in response to acute treatments of RPE1, we found that the OCR after rotenone addition was maximal at 800 nM, whereas oligomycin inhibition of ATPase was maximal already at 200 nM. Of note, these cells display a strong switch towards aerobic glycolysis, which can give an advantage on cell survival during mitochondria impairment (Fig 4C). Using rotenone and oligomycin treatments with the doses that maximally inhibit OCR for 48 h, we found that the number of ciliated cells was unaltered compared with control (Figs 4D and S4F). However, we found again a dose-dependent increase of median (Figs 4E and S4G) and maximum ciliary length (Figs 4F and S4H), using rotenone but not oligomycin.

To investigate whether cellular energy status was responsible for the ciliary changes observed in RPE1 cells, we measured adenosine nucleotides after rotenone or oligomycin treatments. We found that oligomycin decreased significantly the ATP/ADP ratio in comparison with control and rotenone treatment (Figs 4G and S4I), the latter displaying a non-significant reduction compared with control cells.

---

image. Data are represented by median+IQr and range (n = 4). Statistical test ANOVA+Bonferroni. **(F)** Cilia length of collecting ducts cells in wild-type and in db/db mice at 8 and 24 wk of age. Data are represented by median+IQr and range (n = 4). Statistical test ANOVA+Bonferroni. **(G)** Gene expression measured by qPCR in wild-type and db/db mice at 8 and 24 wk of age. Data are represented by mean + SD (n ≥ 3). **(H)** Flux analysis recording of the oxygen consumption rate from complex I+III+IV, II+III+IV, or complex IV only of disrupted mitochondrial from frozen samples extracted from the kidney of wild-type and db/db mice at 8 and 24 wk of age. Data are represented by mean + SEM (n ≥ 3). **(I)** Oxygen consumption rate from complex I+III+IV, II+III+IV, or complex IV of disrupted mitochondrial from frozen samples extracted from kidney of wild-type and db/db mice at 8 and 24 wk of age. Data are represented by mean + SD (n ≥ 3). **(J)** Representative single stack images of kidney sections stained with ciliary acetylated tubulin (green) and distal tubules cells Na⁺-K⁺ ATPase (red). Nuclei (blue) were counterstained with DAPI. Legend: Gray = wild-type mice (wt/wt); black empty = db/db. **(K)** Percentage of ciliated distal tubules cells in wild-type and db/db mice at 8 and 24 wk of age. Data are represented by median+IQr and range (n = 4). Statistical test ANOVA+Bonferroni; statistical significance between wild-type versus db/db: $P < 0.05$ (*), <0.01 (**), or <0.001 (***).

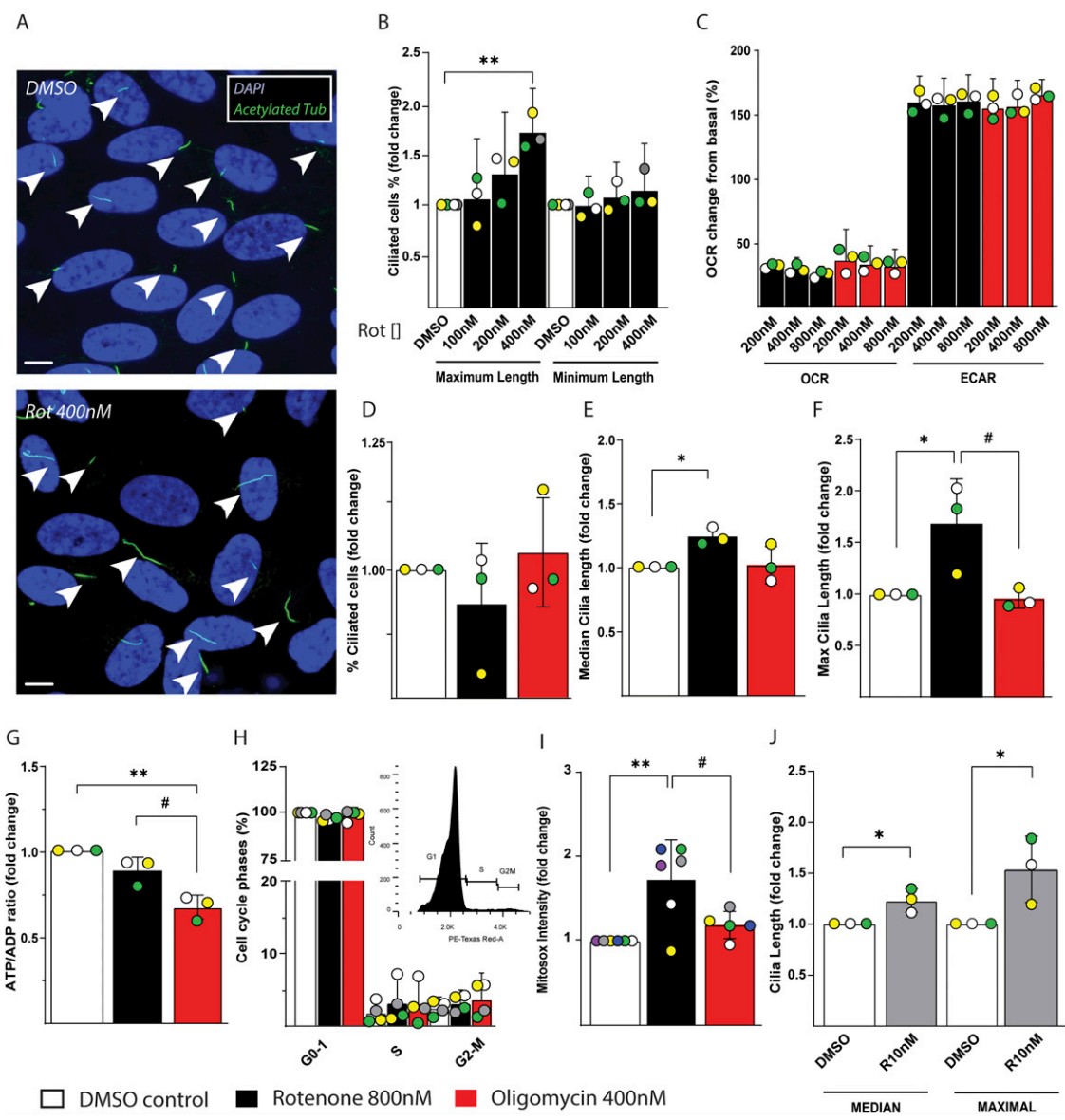

**Figure 4. Rotenone-induced reactive oxygen species production, but not cellular energy status and cell cycle, are responsible for elongating cilia in RPE1 cells.**
**(A)** Representative images of RPE1 cell line treated 48 h with DMSO vehicle or 400 nM rotenone. Cilia stained with anti-acetylated tubulin are indicated with white arrows. Scale bar = 5 µM. **(B)** Quantification of dose-dependent effect of rotenone on maximum and minimum cilia length in RPE1 cells. Data are represented by mean + CI; n = 3. **(C)** Oxygen consumption rate (OCR) and extracellular acidification rate (ECAR) normalized to the baseline after addition of different doses of rotenone and oligomycin in RPE1. Data are represented by mean + CI; n = 3. **(D)** Number of ciliated RPE1 after 48 h of 800 nM rotenone or 400 nM oligomycin versus DMSO vehicle control. Data are represented by mean + SD; n = 3. **(E)** Quantification of the median ciliary length in RPE1 treated as above. Data are represented by mean + SD; n = 3. **(F)** Quantification of the maximum ciliary length in RPE1 treated as above. Data are represented by mean + SD; n = 3. **(G)** ATP/ADP ratio normalized to the DMSO control in RPE1 cells treated 48 h as above. Data are represented by mean + SD; n = 3. **(H)** Cell cycle analysis in RPE1 cells treated as above. In the inset is displayed a representative image of cell cycle plot in control cells. Data are represented by mean + CI; n = 4. **(I)** Quantification of Mitosox intensity in RPE1 cells treated as above. Data are represented by mean + SD; n ≥ 5. **(J)** Quantification of median and maximum ciliary length in RPE1 cells treated with 10 nM rotenone in glucose-free galactose media. Data are represented by mean + SD; n = 3. Legend: Black empty = DMSO control; black = rotenone 800 nM; red = oligomycin 400 nM; gray = rotenone 10 nM. Statistical significance between control versus treatments: $P < 0.05$ (*), $<0.01$ (**). Rotenone versus oligomycin: $P < 0.05$ (#).

This finding excludes the causative role of cellular energy status on ciliary changes in RPE1 in response to rotenone treatment.

Cilia reabsorption and elongation is dependent on the cell cycle (Kasahara & Inagaki, 2021). After rotenone treatment, we did not find changes compared with control and oligomycin treated cells (Figs 4H and S4J). Moreover, after 5 d of starvation most of the RPE1 cells are in

G1 phase, allowing us to exclude the cell cycle as a reason for ciliary morphological changes in response to rotenone treatment.

Taken together, these results show that in RPE1 cells rotenone but not oligomycin induced an elongation of cilia in a dose-dependent manner, without altering cilia number per cell. The lower ATP/ADP ratio in cells treated with oligomycin suggests

that the elongation of cilia due to rotenone treatment is independent of the cellular energetic status.

## ROS are responsible for ciliary morphology alteration in RPE1 cells

Knowing the diversity of cellular functions between RPE1 and IMCD3 cells (Strauss, 2005; Plafker et al, 2012), we next hypothesized that ROS production in RPE1 can trigger a different response compared with IMCD3 cells. In RPE1 cells, we found that rotenone treatment increased mitochondrial ROS generation (Figs 4I and S4K and L) that was not reflected by an increase in cytosolic ROS (Fig S4M and N). The lack of cytosolic ROS as consequence of rotenone and oligomycin treatments suggests the presence of effective ROS detoxification mechanisms. As previously mentioned, we found that ROS were in part responsible for ciliary impairment in IMCD3 and that antioxidant treatment partially restored ciliary morphology. We tested a similar rescue experiment in RPE1 cells; however, antioxidant treatment could not rescue the changes in maximum ciliary length induced by rotenone in this cell model (Fig S4O). Accordingly, blocking radical production induced by rotenone might be difficult in this specific cell type as indicated by a previous study (Lu et al, 2006). This can be due to the antioxidant mechanism of action and radical site in which the ROS are produced and eventually quenched.

In RPE1 cells, off-target effects induced by rotenone other than ROS production would disappear using lower rotenone concentrations. Thus, we cultured the cells in glucose-free galactose media inducing the cells to use almost exclusively mitochondria as energy source. With this method, we reduced the rotenone amount by 80-fold to reproduce similar effect seen in media with glucose. In fact, using a concentration of 10 nM of rotenone for 48 h, we reproduced the increase in median and maximum ciliary length (Figs 4J and S4P) we found using higher rotenone concentrations in glucose media. The similar phenotype was observed using galactose culture in RPE1 treated with lower concentrations of rotenone showed that the elongation of cilia was triggered by mitochondrial ROS. Interestingly, a single injection of the mitochondrial toxin MPTP in mice, which inhibits complex I and induces ROS, has been show to produce elongation of cilia in dopamine neurons (Bae et al, 2019). This highlights the role of mitochondria as an internal radical producer causing cilia elongation in RPE1 cells, which might be similar to the effect seen previously in human fibroblast (Burkhalter et al, 2019) but different from other cell types such as IMCD3.

In conclusion, interfering with cellular energetic status and promoting intracellular ROS from mitochondria induces changes in cilia morphology, which is cell dependent. In vitro findings in collecting duct cells were translated in vivo, where ciliary impairments where found during diabetic nephropathy in a mouse model that closely represents the human progression of the disease. The findings described here warrant future research focusing on ciliary impairments during conditions characterized by intracellular ROS overproduction, such as diabetes and aging, as well as on the molecular mechanisms that links mitochondrial dysfunction and ROS overproduction to primary cilia.

# Materials and Methods

All chemicals were purchased from Sigma-Aldrich unless otherwise stated.

## Cell culture and treatments

*Mouse inner medullary collecting duct cell lines* (IMCD3) were donated by Berggren's (Karolinska Institutet) and Lickert's groups (Helmholtz Zentrum Munich). IMCD3 were grown in DMEM F:12 (Gibco) supplemented with 10% FBS, penicillin 100 UI/ml, and streptomycin 100 $\mu$g/ml. Cells were incubated at 37° and 5% $CO_2$ and passaged before confluence. *Human telomerase reverse transcriptase immortalized retinal pigmented epithelial 1 cell line* (RPE1) was purchased from ATCC. RPE1 were grown in DMEM F:12 (Gibco) supplemented with 10% FBS and 0.01 mg/ml Hygromycin B. For all the experiments during confluence, 20,000 cells/cm$^2$ were plated at day 0 and allowed to reach confluence (day 4). The same day the cells were starved using the above described medium supplemented with 0.5% FBS (starving medium) until day 7. The cells were than treated according to the experiment in starving medium containing rotenone or oligomycin for 48 h. The medium was changed after 24 h to avoid the decrease of pH due to aerobic glycolysis. IMCD3 cells were treated with 25 nM rotenone and 50 nM oligomycin. These concentrations have been chosen because of the cell death with higher doses. The concentrations of these inhibitors for RPE1 cells were chosen evaluating the maximum inhibition of CI and ATPase, respectively. In the experiments using glucose-free galactose medium (DMEM/F12 1:1 [PAN-Biotech Gmbh]) supplemented with 16 mM galactose and 2 mM GlutaMax, the medium was supplemented with dialyzed FBS to avoid the presence of glucose (PAN-Biotech Gmbh). All cells used were tested monthly for mycoplasma contamination.

## Animals and biochemical parameters

Animal experiments were conducted using 8- (prior diabetic nephropathy onset) and 24-wk (during diabetic nephropathy) female BKS.Cg-*Dock7$^{m+/+}$Leprdb*/J (db/db) mice and corresponding age-matched wild-type (wt) littermates (Charles River Laboratories). Animal experiments were performed in accordance with the Animal Experiment Ethics Committee at Karolinska Institutet (ethical permission numbers N445/12 and N199/13).

Non-fasting glucose, triglycerides, creatinine, hormones, and cholesterol levels were measured using blood obtained from the tail vein, prior euthanasia. Blood glucose was determined using FreeStyle Glucometer (Abbot Diabetes Care). Triglycerides and cholesterol were determined by a multi-parameter diagnostic device (multiCare-in, Biochemical Systems International S.r.l.). Creatinine levels were measured in deproteinized plasma by centrifugation in 10 kD spin columns (Abcam) using a commercial colorimetric kit (Abcam). Plasma insulin, c-peptide, glucagon, adiponectin, and leptin were measured using ELISA (CrystalChem and Phoenix Pharmaceuticals Inc. for leptin).

## Generation of IMCD3 stable cell lines

pCag-ARL13b-Venus and pCag-ARL13b-RFP constructs used to generate IMCD3 stable cell line were kindly donated by Dr. Ingo Burtscher (Helmholtz Zentrum Münich). To generate IMCD3 stably expressing mitochondrial photoactivatable GFP (Mito-PAGFP [Karbowski et al, 2004]) and ARL13b-RFP, the ORF was amplified using Phusion hot start II DNA polymerase (Invitrogen) from the original Mito-PAGFP (#23348; Addgene) using primers with end sequences compatible with the pCag-ARL13b-RFP vector. The amplification product was cut, purified (QiaQuick), and ligated with the open vector at 14° overnight followed by 2 h room temperature (rT) using 1 µl T4 ligase (Invitrogen). After purification, 1.6 µg of linearized plasmid was bound with Lipofectamine 2000 and the solution transferred into the 12-well culture plate containing 200,000 cells/well seeded the day before transfection. After 48 h, 1 µg/ml puromycin was added to induce selection.

## Metabolic flux analysis (seahorse)

Metabolic analysis was performed using Seahorse XF96 Extracellular Flux Analyzer (Agilent). 5,000 cells/well RPE1 were seeded and treated as described in cell culture and treatments section. The day of the experiment the growing medium was replaced with XF medium containing 2 mM GlutaMax (Gibco), 1 mM sodium pyruvate, 16 mM glucose, and 0.5% FBS. Cells were incubated at 37°C without $CO_2$ 1 h prior the experiment. Basal OCR following injection of different concentrations of oligomycin, rotenone, and rotenone+antimycin (4 µM final each) was measured. Running template was 2 min mix, 1 min wait, and 4 min measure.

Mitochondria complexes respiration from frozen samples was measured with Seahorse XF96 (Agilent) as previously described with minor modifications (Acin-Perez et al, 2020). Briefly, 10–20 mg of frozen kidney was thawed in ice-cold PBS, minced, and mixed with 500 µl of mitochondria assay solution without BSA (MAS; 220 mM d-Mannitol, 70 mM sucrose, 10 mM of $KH_2PO_4$, 5 mM $MgCl_2$, 2 mM Hepes, and 1 mM EGTA, pH 7.4 at 37°C). The digested tissue was mechanically homogenized with 10 strokes of Dounce homogenizer. To each sample, a further 500 µl of MAS with BSA 0.2% (wt/vol) of fatty acid-free BSA 0.2% (wt/vol) in ultrapure $H_2O$, pH 7.2 at 37°C, was added and the samples centrifuged 1,000$g$ 10 min at 4°C. 8 µg of protein homogenate was loaded in a XF microplate in 20 µl MAS and centrifuged 2,000$g$ 5 min at 4°C (no brake). For each well, 130 µl of pre-warmed (37°C) MAS with ADP (final concentration 4 mM) was added and the plate placed at 37°C in a non-$CO_2$ incubator for the time of cartridge calibration. The OCR was measured at basal level, after addition in MAS with NADH (to measure activity of CI+III+IV) (1 mM) or succinate (5 mM)+rotenone (2 µM) (to measure activity of CII+III+IV), rotenone+antimycin to inhibit complexes I and III, ascorbate (1 mM) + TMPD (0.5 mM) to measure complex VI activity and finally sodium azide (50 mM) to inhibit complex IV. The respiration was measured using cycles of 1 min mix and 5 min measure.

## Adenosine nucleotides content

Cells were washed once in PBS and ice-cold perchloric acid 1M was added. The plates were immediately put on dry ice and stored at –80°. The cells were scraped on ice and the suspension centrifuged at 14,000$g$ at 4° 10 min. The supernatant was neutralized using KOH 20M and phosphate buffer (1.7M $KH_2PO_4$ and 0.24M $K_2HPO_4$, pH 7) and stored at –80° for 30 min. The solution was thawed on ice and centrifuged as before. The supernatant was collected and 50 µl were injected in the HPLC. The separation module used was a Waters 2795 (Waters) equipped with Supelcosil LC-18 15 cm × 4.6 mm/3 µm column, guard column, and Waters 2996 photodiode array detector. The filtered (45 µm) mobile phase used for separation consisted of Buffer A (23.3 mM $KH_2PO_4$ + 1.7 mM $K_2HPO_4$ + 10 mM tetrabuthylammonium sulfate pH 5.7) and B (Methanol HPLC grade Promochem). The linear gradient used for separation at 0.7 ml/min was from A 91.7% B 8.3% until B 27.7% at 24 min and then immediately B 8.3% until 32 min. The day of the experiment the column was equilibrated with 50 column volumes of buffer A and at the end of the day the column was flushed with 10 volumes of water and 10 volumes of water:methanol 70:30. The peaks were analyzed using Mass Lynx spectrometry software (Waters). The amount of adenosine nucleotides were calculated based on standard curve performed using 100 mM adenosine nucleotides standards.

## Mitochondrial intermembrane potential

We seeded 10,000 IMCD3 cells in IBID eight-well chamber allowed to reach confluence under basal culture conditions as described in cell culture and treatments section. The IMCD3 ARL13b-Venus cell line was loaded with TMRE (Molecular probes) at a final concentration of 25 nM in the medium 1 h prior the measurements. The measurements were performed using a LEICA SP5 II confocal microscope (Leica), with heating plate station at 37° and 5% $CO_2$. Microscope settings: objective 63× zoom 2, 4% laser power to avoid photobleaching and photodamage, 512 × 512 pixel, ex/em TMRE 514/565–600, pinhole 1 Airy. Z-stacks 0.25 µm. Mitochondria as surface objects were selected using the surface tool of Imaris software. Using the mean fluorescence per object and the volume of the object we calculated the mean fluorescence/voxel of all cellular mitochondrial network and the mitochondria within 2 µM distance from each primary cilium (Fig S1B). Every image included at least 5–10 cells.

## Ciliary and mitochondria staining in vitro

20,000 IMCD3 or RPE1 cells were seeded onto 12 mm diameter glass coverslips in a 24-well plate and treated as described in cell culture and treatments sections. On the day of the experiment, the cells were washed once with PBS and fixed using 3% paraformaldehyde in PBS at 37° for 10 min. The cells were washed three times in PBS and incubated with blocking and permeabilization solution (5% FBS in PBS 0.1% Triton X-100) for 30 min. After washing once, the cells were incubated overnight at 4° with primary antibodies against ciliary localized proteins acetylated tubulin (mouse) and ARL13b (rabbit) (Table S1). The day after, the wells were washed three times with PBS and fluorescent secondary antibodies donkey anti-mouse Alexa 488 and donkey anti-rabbit Alexa 555 (Table S1) were applied for 1 h at rT. After the first wash (10 min) with PBS containing 1 µg/ml DAPI, the coverslips were washed twice and mounted. LEICA SP5 II confocal microscope equipped with Hybrid detectors was used for

imaging. Microscope settings: objective 63×, 512 × 512, sequential scanning, pinhole 1 Airy. Zoom, gain, and laser power were dependent on the assays. For measurement of ciliary morphology in RPE1 cells, z-stack were optimized to avoid loss of information. In RPE1, cilia length was measured using the CC bounding box tool in Imaris, which identifies the maximum length for the long axis of the object. Cilia number was quantified by software recognition of number of objects in RPE1 cells or by visual counting in IMCD3 because of the presence of cytokinetic bridges and other acetylated tubulin structures in these cells. For each technical replicate, between 50 and 100 cells were scored for cilia number and/or length.

## Western blot

### IMCD3
Cells were washed in PBS and ice-cold RIPA-modified lysis buffer (150 mM NaCl, 50 mM Tris–HCl, pH 7.4, 1% NP-40, 1 mM EDTA, and 0.1% DOC) supplemented with Pierce phosphatase and protease inhibitor (Thermo Fisher Scientific). The suspension was transferred in a microfuge tube on ice and vortexed sometimes for 30 min. The tubes were centrifuged at 4° 15 min 18,000$g$ and the supernatant stored at –80° until usage. After dilution in loading buffer and reducing agent DTT, the samples were boiled for 10 min to allow complete denaturation of the proteins. The membranes were blocked for 1 h at RT using Odyssey blocking buffer (Li-Cor biosciences). The membranes were washed three times for 10 min in TBST and incubated overnight with the appropriate antibodies in Odyssey blocking buffer. The membranes were washed again and incubated 1 h in the dark using fluorescent secondary antibodies (Li-Cor). The list of antibodies used is reported in Table S1.

## mRNA processing and qPCR

After euthanasia, part of one kidney was quickly dissected and submerged in RNAlater (Thermo Fisher Scientific), incubated at 4°C for 24 h and the excess of liquid decanted, and the tissues stored at –80°C. 20–30 mg were used to extract mRNA using RNase plus mini kit (QIAGEN) was used. Extracted RNA concentration and purity was measured using a nanophotometer (Implen GmbH). For cDNA synthesis, 50–1,000 ng of RNA was processed with Maxima First Strand cDNA Synthesis Kit (Thermo Fisher Scientific), following the manufacturer's instruction. For the qPCR experiments, 1 $\mu$l of cDNA or plasmid templates were loaded into a 384-well plate together with 1 $\mu$l of 10 $\mu$M primer pairs, 3 $\mu$l H$_2$O, and 5 $\mu$l SYBR Green PCR Master Mix (Thermo Fisher Scientific) and run using QuantStudio5 thermocycler (Thermo Fisher Scientific). The primers used are listed in Table S2. For every assay, a reverse transcriptase negative control (–rt) with all primers used was run. Three reference genes (*hmbs*, *tbp*, and *hprt*) were used to calculate the geometric mean and this was used to normalize the gene expression.

## Tissue immunology and immunohistochemistry

Animals were anesthetized with isoflurane and perfused with PBS followed by 4% (wt/vol) paraformaldehyde in PBS. Prior cryo-preservation kidneys were processed with a sucrose gradient (10–30% wt/vol sucrose in PBS containing 0.01% sodium azide and 0.02% bacitracin), frozen in dry ice, and preserved at –80° prior usage. 20-$\mu$m sections were counterstained with hematoxylin–eosin for morphological studies.

For immunohistochemistry, 20-$\mu$m-thick longitudinal cryosections were collected on Superfrost Plus slides. After sectioning, the slides were equilibrated for 2 h at rT and washed once for 5 min in PBS. Tissue permeabilization was performed using 0.5% (vol/vol) Triton X-100 in PBS for 15 min. Thereafter, blocking solution containing 5% (wt/vol) BSA and 0.5% (vol/vol) Triton X-100 in PBS was applied for 1 h at rT before overnight incubation at 4° with primary antibodies (Table S1). The sections were washed three times for 5 min with 0.05% (vol/vol) Tween-20 in Tris/NaCl (0.1M Tris-Base; 0.15M NaCl). For double labeling, the corresponding secondary antibodies (donkey anti-mouse Alexa Fluor 488, 1:500 and goat anti-rabbit Alexa Fluor 594) were applied at rT for 1 h. After incubation, sections were washed again and mounted using a mounting medium containing DAPI. Images were captured using a Leica SP5 II confocal microscopy as described above with 63× objective, zoom 3.0, 0.6 $\mu$m stack distance and 1,024 × 1,024 resolution. Cilia number and length were measured in the collecting duct and distal tubule cells using two non-consecutive kidney sections separated by 200 $\mu$m. For each section, three fields were collected in four individual animals per experimental group and controls at both ages. Cilia number and length were analyzed in double blind using ImageJ software.

## ROS measurements and cell cycle analysis by flow cytometry

To detect cytosolic ROS carboxy-methyl-2'-7'-dichlorofluorescin diacetate (CM-H2DCF-DA; Molecular Probes) was loaded at final concentration of 10 $\mu$M for IMCD3 and 2.5 $\mu$M for RPE1 in the culture medium for 30 min. To detect mitochondrial superoxide we used 2.5 $\mu$M Mitosox Red (Molecular Probes) for 1 h. After the incubation time, cells were harvested, washed, and resuspended in PBS, and fluorescence intensity was measured.

To measure apoptosis in living cells we used Annexin V-FITC Apoptosis Detection Kit (Abcam), according to the manufacturer's instruction. Briefly, IMCD3 cells were trypsinized, washed in medium and resuspended in 500 $\mu$l 1× binding buffer containing 5 $\mu$l Annexin V FITC and 0.5 $\mu$g/ml Propidium Iodide (PI). The tubes were incubated in the dark for 5 min and put on ice before measurements. Negative control with non-stained cells and cells stained only with Annexin V FITC or PI were used to set the gate. In the resulting graph plotted with FITC versus PE-Texas red we identified a double negative population (living non-apoptotic cells), a positive for Annexin V or PI (apoptotic and dead cells, respectively) and double positive population (apoptotic dead cells).

To measure cell cycle, cells were trypsinized, washed in PBS, and fixed with 500 $\mu$l 70% precooled ethanol at –20°. After storage on ice for 1 h, tubes were centrifuged 2 min at 1,700$g$. The pellet was resuspended in 500 $\mu$l PBS and returned to ice for 30 min followed by another centrifugation step. The pellet was resuspended in 300 $\mu$l PBS containing 10 $\mu$g/ml RNase A, 0.25% Triton X-100, and 20 $\mu$g/ml PI. The solution was transferred in FACS tubes and incubated in the dark for 15 min and then on ice prior measurements. Plotting the forward scatter versus the PE-Texas signal we analyzed the typical distribution represented by two peaks (G1 and G2/M phases)

and a valley (S phase). The analyzed peaks were reported as phase's percentage of the total cells analyzed.

Flow cytometric experiments were performed using BD FACSaria III (BD Biosciences). Negative control for each experiment was performed in not stained cells. The data were collected using minimum 20,000 cells and analyzed using FLOWJO software. The cell population of interest was identified using forward and side scatters and doublet discrimination (scatter width versus area).

## Statistics

Data representation, graphics, and statistics have been performed using GraphPad Prism 8.3 software after outliers' removal. Statistics on biological replicates shown in the main figures have been performed using ANOVA combined with Tukey's post hoc test unless otherwise stated. Data are represented as mean + SD normalized on the control of that experiment (thus the controls are always set as 1). Statistics including all technical replicates shown in supplementary figures has been performed depending on the statistical distribution of the data and number of groups. ANOVA, $t$ test, or Kruskal-Wallis combined with Tukey or Dunn's post hoc tests were used. Data are represented as mean + CI unless otherwise stated. For ciliary length measurements, every experiment, which included three to five technical replicates, was normalized to the DMSO control of that experiment and the length expressed as fold change compared with control. Each experimental set was replicated at least three times unless otherwise stated. Statistical significance was considered for $P < 0.05$ (*), $<0.01$ (**), or $<0.001$ (***).

## Data Availability

The data that support the findings of this study are available from the corresponding author, upon reasonable request.

## Supplementary Information

## Acknowledgements

We would like to thank all members of the Professor Lickert's lab and the Institute for Diabetes and Regeneration Research at Helmholtz Munich and especially Dr. Jantje Gerdes for their helpful suggestions, technical expertise, and discussions. This work was supported by Family Erling-Persson Foundation, Medical Research Council D0284901, Funds of Karolinska Institutet.

## Author Contributions

N Moruzzi: conceptualization, data curation, formal analysis, validation, investigation, visualization, methodology, and writing—original draft, review, and editing.
I Valladolid-Acebes: formal analysis, validation, investigation, methodology, and writing—review and editing.
SA Kannabiran: investigation, methodology, and writing—review and editing.
S Bulgaro: investigation, visualization, methodology, and writing—review and editing.
I Burtscher: resources, methodology, and writing—review and editing.
B Leibiger: validation, methodology, and writing—review and editing.
IB Leibiger: validation, methodology, and writing—review and editing.
P-O Berggren: resources, funding acquisition, visualization, and writing—review and editing.
K Brismar: resources, supervision, funding acquisition, visualization, project administration, and writing—review and editing.

## Conflict of Interest Statement

P-O Berggren is the cofounder and CEO of Biocrine AB; IB Leibiger and B Leibiger are consultants for Biocrine AB.

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
