## [Reviewer comments · Life Science Alliance]

Life Science Alliance

MITOCHONDRIAL IMPAIRMENT AND INTRACELLULAR REACTIVE OXYGEN SPECIES ALTER PRIMARY CILIA MORPHOLOGY

Noah Moruzzi, Ismael Valladolid-Acebes, Sukanya Kannabiran, Sara Bulgaro, Ingo Burtscher, Barbara Leibiger, Ingo Leibiger, Per-Olof Berggren, and Kerstin Brismar

DOI: <https://doi.org/10.26508/lsa.202201505>

Corresponding author(s): Noah Moruzzi, Karolinska Institute

Review Timeline:

Submission Date:	2022-04-26
Editorial Decision:	2022-06-06
Revision Received:	2022-07-29
Editorial Decision:	2022-08-29
Revision Received:	2022-09-02
Accepted:	2022-09-05

Transaction Report:

June 6, 2022

Re: Life Science Alliance manuscript #LSA-2022-01505-T

Dr. Noah Moruzzi
Karolinska Institute
Molecular medicine and surgery
Anna Steckséns Gata 53
Solna 17176
Sweden

Dear Dr. Moruzzi,

Thank you for submitting your manuscript entitled "MITOCHONDRIAL IMPAIRMENT AND INTRACELLULAR REACTIVE OXYGEN SPECIES ALTER PRIMARY CILIA MORPHOLOGY" to Life Science Alliance. The manuscript was assessed by expert reviewers, whose comments are appended to this letter. We invite you to submit a revised manuscript addressing the Reviewer comments.

Thank you for this interesting contribution to Life Science Alliance. We are looking forward to receiving your revised manuscript.

Sincerely,

B. MANUSCRIPT ORGANIZATION AND FORMATTING:

Reviewer #1 (Comments to the Authors (Required)):

In this manuscript, Moruzzi et al. examine the role of mitochondrial dysfunction on cilia length control in two different immortalized cell lines, IMCD3 kidney epithelial cells and RPE retinal pigment epithelial cells. Using chemical inhibitors of electron transport protein complexes I and IV, they examine whether mitochondrial oxidative stress and/or reduced ATP generation affects cilia length. They show that the regulation of cilia phenotype by mitochondria is cell type specific. In IMCD3 cells, rotenone and oligomycin reduce percentage of ciliated cells, indicating that both intramitochondrial oxidative stress and reduced ATP contribute to this. In contrast, in RPE1 cells, rotenone only increases cilia length, indicating that intramitochondrial oxidative stress, but not cellular energy, affects cilia length and in a converse fashion from IMCD3 cells. Further, to demonstrate that mitochondrial dysfunction affects ciliogenesis in vivo, the authors show shortened cilia lengths in medullary collecting duct cells of db/db mice, a model of diabetic nephropathy.

The investigation of the role of mitochondria on cilia biology and the finding of cell type differences are important. Using inhibitors of different electron transport protein complexes to tease out different mitochondrial roles is another strength. A weakness is that this reviewer does not understand why the authors chose to look at db/db collecting ducts, since diabetic nephropathy typically affects podocytes and glomeruli. Additionally, characterization of the kidney phenotype in db/db mice eg. mitochondrial phenotype in the various renal tubular segments, fibrosis, glomerular damage to correlate with cilia phenotypes is missing.

Major Concerns:

1. Figure 1. Are cilia lengths also reduced? Please include cilia lengths.
2. Figure 2. Since methods come after the results/discussion, please state what Mitosox measures. Similarly, define DCFDA and what it measures.
3. Figure 2. Please also include cilia lengths. Do the antioxidants also rescue potential cilia length phenotype?
4. Figure 3. In general, it is unclear why the medullary collecting ducts of db/db mice are important to the disease. The authors seem to want to correlate what is seen in IMCD3 cells in vivo, but please explain how these experiments are relevant. Do the cilia lengths in the medullary collecting ducts occur after the mitochondrial defects? If so, please show the mitochondrial defects in these collecting ducts at 8wks and 24wks.
5. Figure 3. Please indicate whether mice are male or female.
6. Supplementary Figure 2. Please show individual data points and indicate whether mice are male or female.
7. Figure 4. Please explain how data of maximum, median and minimum cilia lengths were decided upon? Usually, all cilia lengths are placed onto same graph.

Minor Concerns:

1. p. 4 - change especial to special
2. p. 5, 1st sentence - change though to thought
3. Figures 4F, 4J; Supplementary Figure 3A - change length to length

Reviewer #2 (Comments to the Authors (Required)):

The manuscript by Moruzzi et al., aims to investigate the relationship between mitochondrial activity and primary cilia, and the role reactive oxygen species play regulating primary cilia. Although the overarching aim of the work is novel and interesting, the conclusions that are drawn are, for the most part, not supported by the data. Moreover, most of the findings have previously been reported and it is therefore unclear what advances in knowledge the manuscript provides. Furthermore, in the results/discussion section there is a lack of technical information and detail that makes it 1) difficult to interpret the data and 2) reading the manuscript disjointed.

Major comments:

Finding 1 (Figures 1A-E): Mitochondria are in close proximity to the primary cilia - Not Supported.

1. The authors report a "distinct subset of mitochondria in proximity to the cilium", however, to this reviewer that was not apparent in the images presented. Furthermore, how did the authors define "proximity to the cilium", it states in the legend <2 um, but does not provide any other information. Given the broad subcellular localization of the mitochondria, have the authors considered random coincidence underpinning the reported apparent proximity of the mitochondria and cilia? In the images presented there does not appear to be specific enrichment of the mitochondria at the cilia.

2. The authors use IMCD3 cells stably expressing ARL13B, however, no control cells are presented. The over- and under-expression of ARL13B has been shown to affect the subcellular localization of proteins including ciliary proteins, and its expression in the IMCD3 cells may impact the distribution of mitochondria. Have the authors performed these studies in control IMCD3 cells expressing vector control?

The figures would benefit from labels including the stains/antibodies/constructs presented. It is confusing because the organelle markers are different colors in panels of the same figure. Can they be pseudo-colored the same for consistency?

Finding 2 (Figures 1F-J): Blocking complex I (CI) or ATPase decrease energy levels and ciliation in IMCD3

1. The authors report fewer ciliated cells upon treatment of IMCD3 cells with Rotenone and Oligomycin. However, how the ciliated cells were identified/scored is not reported in the main text, requiring the reader to go to the legends which are brief.

Was an axoneme and basal body marker used to identify cilia? During the final stages of cytokinesis a cytokinetic bridge is present between the daughter cells and may be mistaken as cilia which is part of the reason why a basal body marker is important. In fact, cytokinetic bridges are apparent in some of the images presented.

2. It appears the authors used acetylated tubulin antibodies to identify cilia in IMCD3 cells. Why was this approach used as the cells are seemingly already expressing ARL13B that can be used to identify cilia?

3. The authors claim a reduced "level" of IFT88 but it is unclear how this was determined ie. protein expression by Western blot, immunofluorescence etc, and whether the reduced IFT88 is specifically at cilia or at the whole cell level. Moreover, was the level of IFT88 measured relative to ARL13B or ac-tubulin?

4. Why are "% ciliated cells" presented as a fold change rather than the absolute numbers? This reviewer is surprised by the apparently high statistical difference/low p-values for all of the graphs in Fig 1, particularly given the broad spread of the data points. How did the authors define "n"? How many cells were scored, and how many biological and technical replicates were undertaken? What statistical tests were done on the data?

5. The authors conclude that Rotenone and to a lesser extent Oligomycin impair ciliogenesis. However, from the methods it appears the classical ciliogenesis assay was not undertaken and instead a modified version of a cilia disassembly assay was done. Why do authors conclude the effect of Rotenone and Oligomycin is on ciliogenesis rather than promoting cilia disassembly? Have the authors attempted to discriminate between the two processes?

Finding 3 (Figure 2): Reactive oxygen species (ROS) are partly responsible for decreased number of ciliated IMCD3 cells - Not supported

Have the authors considered that Rotenone has several intracellular actions including the depolymerization of tubulin and that its other actions may contribute to the regulation of cilia? How specific are the functions of the rescue agents such as Tiron?

How was the cell scoring and statistics done?

Finding (Figure 3): Ciliation is impaired in the kidney of a diabetic nephropathy mouse model.

The authors use db/db mice which display a similar phenotype to diabetic nephropathy in humans, and report impaired ciliation in the kidney. However, apart from noting that diabetic nephropathy kidneys "are subjected to increased oxidative stress and mitochondrial dysfunction" there is no evidence provided to demonstrate a causative association between oxidative stress/mitochondrial dysfunction and the cilia defects. Furthermore, cilia defects in various tissues of db/db mice have previously been reported.

Perhaps the authors could consider doing "rescue" studies using primary db/db cells to investigate the role of mitochondria and ROS in the regulation of cilia.

Finding (Figure 4): Mitochondrial blockade elongates cilia in RPE cells independently of cellular energy status/ROS are responsible for ciliary morphology alteration in RPE cells

The authors report rotenone treatment (but not oligomycin) of RPE cells increased cilia length. Previous studies have reported the similar observations in RPE cells as well as other cell types such as neurons. Therefore, it is not clear what the new findings are here. Furthermore, I am surprised at the level of statistical significance for the graphs in Figure 4 and would like clearer descriptions of the analysis.

Reviewer #1 (Comments to the Authors (Required)):

In this manuscript, Moruzzi et al. examine the role of mitochondrial dysfunction on cilia length control in two different immortalized cell lines, IMCD3 kidney epithelial cells and RPE retinal pigment epithelial cells. Using chemical inhibitors of electron transport protein complexes I and IV, they examine whether mitochondrial oxidative stress and/or reduced ATP generation affects cilia length. They show that the regulation of cilia phenotype by mitochondria is cell type specific. In IMCD3 cells, rotenone and oligomycin reduce percentage of ciliated cells, indicating that both intramitochondrial oxidative stress and reduced ATP contribute to this. In contrast, in RPE1 cells, rotenone only increases cilia length, indicating that intramitochondrial oxidative stress, but not cellular energy, affects cilia length and in a converse fashion from IMCD3 cells. Further, to demonstrate that mitochondrial dysfunction affects ciliogenesis in vivo, the authors show shortened cilia lengths in medullary collecting duct cells of db/db mice, a model of diabetic nephropathy.

The investigation of the role of mitochondria on cilia biology and the finding of cell type differences are important. Using inhibitors of different electron transport protein complexes to tease out different mitochondrial roles is another strength. A weakness is that this reviewer does not understand why the authors chose to look at db/db collecting ducts, since diabetic nephropathy typically affects podocytes and glomeruli. Additionally, characterization of the kidney phenotype in db/db mice eg. mitochondrial phenotype in the various renal tubular segments, fibrosis, glomerular damage to correlate with cilia phenotypes is missing.

We thank the reviewers for the valuable comments and suggestions, and for the recognition of our efforts. In the point-by-point response, we addressed the concerns raised by the reviewer and add some data, which we think will clarify the point raised.

Major Concerns:

1. Figure 1. Are cilia lengths also reduced? Please include cilia lengths. Unfortunately, we cannot conclude if the cilia length can also be reduced by chemical treatments in this cell type. We did not, on purpose, quantify and therefore include the cilia length of IMCD3 in the manuscript due to several reasons listed here:

- Software-based cilia recognition is not feasible in IMCD3 due to: 1. IMCD3 are dividing cells (contrarily to serum starved RPE1 cell line) and thus acetylated tubulin is present in the cytokinetic bridges (see red circle in the picture below) and thus the length calculated by the software will be incorrect. 2. Acetylated tubulin is present in many cells outside the cilia and depends on the growth condition. Therefore also these other structures will be picked up by the software (see yellow circle in the picture to the right). 3. Co-staining with ARL13b (as shown in the picture to the right) is also not feasible due to the aspecific binding of the antibody outside the cilia in IMCD3 plus the fact that some cilia are ARL13b negative but acetylated tubulin positive (see white circles).
- Depending on the cell confluence, the cilia are shorter or longer in IMCD3 even in the same samples (see the 2 pictures below from the same sample but in two different fields of view).

Double staining of cilia in IMCD3 with antibodies against ARL13b (red) and acetylated tubulin (green). Yellow color indicate double labeling. White circles indicate cilia which are negative for ARL13B but positive for acetylated tubulin. Yellow circle indicates the aspecific staining with anti-acetylated tubulin antibody.

Cilia staining in IMCD3 from two fields of view of the same sample with antibody against acetylated tubulin (green). The red circle indicates cytokinetic bridge not included in the cilia count.

At the beginning of the project, we tried to evaluate the length of the cilia in IMCD3 but due to these technical difficulties and big variation within the same sample, we decided not to continue along this research line. In fact, the quantification of cilia length would be for us misleading and difficult to interpret. For this reason (as described in the method section), we did not optimize the imaging experiments (optimal z resolution) to calculate accurately the cilia length.

2. Figure 2. Since methods come after the results/discussion, please state what Mitosox measures. Similarly, define DCFDA and what it measures. A brief explanatory sentence has been added in the results/discussion to introduce the two dyes.

“By measuring mitochondrial superoxide and cytosolic ROS using MitoSOX and DCFDA, respectively, we found that only rotenone increased the intramitochondrial radicals in IMCD3 (Fig 2A and Fig S1C), but both inhibitors increased the cytosolic radicals to a similar extent (Fig 2B and Fig S1D).”

3. Figure 2. Please also include cilia lengths. Do the antioxidants also rescue potential cilia length phenotype? Please refer to point 1.

4. Figure 3. In general, it is unclear why the medullary collecting ducts of db/db mice are important to the disease. The authors seem to want to correlate what is seen in IMCD3 cells in vivo, but please explain how these experiments are relevant. As pointed out by the reviewer, our aim with these experiments was to translate our finding in IMCD3 to an in vivo model and not an attempt to connect the disease progression to tubular cells. Obviously, glomeruli and podocytes are primary actors in the development of diabetic nephropathy as described in the literature. However, also the tubules, which has been described having the highest mitochondrial density, are clearly involved in the deterioration of the kidney during chronic kidney disease and ROS/RNS are major contributors to the pathology (1). We added these concepts in the text:

“To translate our findings of decrease in ciliation in IMCD3 cells in vitro, to the corresponding type of cell (inner medullary collecting duct cell) in the living organism, we took advantage of a diabetic mouse model namely Leptin-receptor deficient (db/db) mouse”

“Tubular cells have been shown to have one of the highest mitochondrial densities, which is needed to generate a great amount of ATP for the active transport of molecules. In the kidney, and in particular in tubule cells, ROS accumulation has been shown to contribute to chronic kidney disease (Doke & Susztak 2022).”

Do the cilia lengths in the medullary collecting ducts occur after the mitochondrial defects? If so, please show the mitochondrial defects in these collecting ducts at 8wks and 24wks.

The cause/consequence relationship between mitochondrial dysfunction and/or oxidative stress and ciliation defects is a good point, which is very challenging to study in vivo in a specific cell type during disease progression. We want to clarify that not only scarcity in energy production (which can be related to mitochondrial dysfunction in our in vitro experiments) but also oxidative stress can cause decrease in cilia number in IMCD3 and could be translated to collecting duct cells in vivo. We highlight this in the manuscript:

“In line with our hypothesis that mitochondrial defects and/or excessive ROS alters cilia in IMCD, we found that only 24 weeks of age....”

Therefore, to address the point raised by the referee, we measured mitochondrial complexes activity and oxidative stress markers in the kidney of this animal model of diabetic nephropathy.

We found a clear increase in oxidative stress gene markers at 24 weeks indicating that the kidneys of this animal model suffer oxidative stress as previously reported (see references in the manuscript). At 8 weeks of age, we found an increase in two antioxidant genes, which could indicate the beginning of a cellular response to oxidative stress. Measuring complex activities in isolated membranes from the same frozen samples in which we measured cilia length, we found a tendency to increased oxygen consumption from complexes I+III+IV and II+III+IV at 8 weeks. However, at 24 weeks of age the complex activity is fairly similar. Although this would not suggest a mitochondrial defect, it seems that mitochondrial complexes activity worsen at 24 weeks of age compared to 8 weeks in db/db mice. The possible increase of mitochondrial activity at 8 weeks of age, which could trigger increased mitochondrial ROS production, could be due to the hyperfiltration shown in these animals during disease development (2) or due to the amount of substrates (hyperglycemia, hyperlipidemia) which could trigger increase in ROS due to excess of energy (see pathogenesis of diabetic complication (3)).

With these experiments, we can conclude that at least oxidative stress was present at 24 weeks when cilia morphology was impaired, while these alterations were not present at 8 weeks where the cilia morphology was similar to the controls. Thus, we correlated at least ROS production with impaired ciliation in vivo in the db/db kidney; while it is not clear how defective the mitochondria are and how much energy production is impaired, although we found a deterioration in some mitochondrial parameters at 24 weeks compared to db/db 8 weeks old.

After the reviewer comments and the experiments performed we completely revised the related section in the manuscript.

5. Figure 3. Please indicate whether mice are male or female. The mice used were all females and this information is present in the method section:

“Animal experiments were conducted using 8 (prior diabetic nephropathy onset) and 24 weeks (during diabetic nephropathy) female BKS.Cg-Dock7m+/+Leprdb/J (db/db) mice”

6. Supplementary Figure 2. Please show individual data points and indicate whether mice are male or female. The individual data points are now shown in the graphs and the sex further indicated in the figure legend.

7. Figure 4. Please explain how data of maximum, median and minimum cilia lengths were decided upon? Usually, all cilia lengths are placed onto same graph.

For each experiment performed on cells, the ciliary length of that particular experiment was normalized to the control of that experiment (technical replicates). The experiments were repeated 3 times (“biological replicates”) unless otherwise stated. The mean values of minimum, maximum and median length of the control samples are therefore always set to one (plus the standard deviation of the replicates in the same experiment) and do not correspond to the real length which for RPE1 was between 1-10 μ M. This was done because of the variation of cilia length during cell passaging and thus the spread would not show the correct biological finding.

Since in RPE1 cells the minimum cilia length was not changed upon chemical treatment (Fig 4B) it was not represented and quantified in the other experiments. The analysis of the cilia on RPE1 was performed using Imaris software using bounding box CC on the longest axis of the cilium. The length of the longest cilium in the sample was used as maximum cilia length, while to understand if there was a general increase in length we used the median of all cilia length.

We added an explanatory sentence in the ciliary staining and statistical analysis in the method section.

“In RPE1, cilia length was measured using the CC bounding box tool in Imaris, which identifies the maximum length for the long axis of the object. Cilia number was quantified by software recognition of number of objects in RPE1 cells or by visual counting in IMCD3 due to the presence of cytokinetic bridges and other acetylated tubulin structures in these cells. For each technical replicate, between 50-100 cells were scored for cilia number and/or length.”

“For ciliary length measurements, every experiment, which included 3-5 technical replicates, was normalized to the DMSO control of that experiment and the length expressed as fold change compared to control. Each experimental set was replicated at least three times unless otherwise stated.”

Minor Concerns: The changes were made according to the suggestions.

1. p. 4 - change especial to special.
2. p. 5, 1st sentence - change though to thought

3. Figures 4F, 4J; Supplementary Figure 3A - change length to length

Reviewer #2 (Comments to the Authors (Required)):

The manuscript by Moruzzi et al., aims to investigate the relationship between mitochondrial activity and primary cilia, and the role reactive oxygen species play regulating primary cilia. Although the overarching aim of the work is novel and interesting, the conclusions that are drawn are, for the most part, not supported by the data. Moreover, most of the findings have previously been reported and it is therefore unclear what advances in knowledge the manuscript provides. Furthermore, in the results/discussion section there is a lack of technical information and detail that makes it 1) difficult to interpret the data and 2) reading the manuscript disjointed.

We thank the reviewer for the comments and understand that some modification should be made to clarify the novelty of our work, advance in knowledge and technical details. We think that by addressing the valuable comments of the reviewer have improved our work. Please see below the point-by-point response to the reviewer's concerns.

Major comments:

Finding 1 (Figures 1A-E): Mitochondria are in close proximity to the primary cilia - Not Supported.

1. The authors report a "distinct subset of mitochondria in proximity to the cilium", however, to this reviewer that was not apparent in the images presented. We revised figure 1 and supplementary figure 1 to be more explanatory. To do that we highlighted the arbitrary radius (set as $2\mu\text{M}$) in which the mitochondria are in close proximity to the cilium. We also showed with stacked images frame by frame the distances between different mitochondria and base of the cilia (identified by ARL13b). We would like to clarify that in the 3D rendering example in Supplementary figure 1B the z plane is elongated and the green mitochondria ($>2\mu\text{M}$ distance) are placed below the yellow ones ($<2\mu\text{M}$) as now shown in the figure 1. We think these additional pictures clarify the subset of mitochondria closer to the cilium from the rest of the network which is mainly around the nucleus in IMCD3 and far more distant to the cilium (Fig 1A,B).

Furthermore, how did the authors define "proximity to the cilium", it states in the legend $<2\text{ }\mu\text{m}$, but does not provide any other information. Given the broad subcellular localization of the mitochondria, have the authors considered random coincidence underpinning the reported apparent proximity of the mitochondria and cilia? The $2\mu\text{M}$ range was an arbitrary range we chose. In our experiments, we observed the majority of the mitochondria above the cell nuclei, which are disjointed from the cytosolic network (wrapped beside and below the nuclei) and closer to the cilium, within this range. This arbitrary cut-off was used to uncover if there was different mitochondrial bioenergetics in proximity to the cilium where maybe more energy was needed. Our conclusion supports that the mitochondria in this proximity do not have different bioenergetics properties, which is a negative finding but still important in our view to characterize the relationship between these two organelles. For this reason, we did not investigate further this aspect. To be clearer and transparent we modified the paragraph in the Result/discussion section:

"Since within a distance of $2\text{ }\mu\text{m}$ away from the ciliary base (identified with ARL13b) we observed the majority of these mitochondria disjointed from the perinuclear network, we set this distance as arbitrary threshold to compare the Ψm of the mitochondrial subset in proximity to the cilium to the average Ψm of all cellular mitochondria."

In the images presented there does not appear to be specific enrichment of the mitochondria at the cilia. We do not think there is an enrichment of mitochondria close to the cilium, but only observed a subset of mitochondria, which is spatially detached from the rest that is commonly around the nucleus. An increase in number of mitochondria, which we did not notice, must have been quantified to state that we have an enrichment close to the cilium. In the text we stated: “However, we observed a distinct subset of mitochondria in proximity to the cilium that was disconnected from the majority of the cell mitochondrial network (Fig 1A,B and Fig S1A). “

2. The authors use IMCD3 cells stably expressing ARL13b, however, no control cells are presented. The over- and under-expression of ARL13b has been shown to affect the subcellular localization of proteins including ciliary proteins, and its expression in the IMCD3 cells may impact the distribution of mitochondria. Have the authors performed these studies in control IMCD3 cells expressing vector control? No, we did not perform studies using a control transfection. The reason was these ARL13b-tagged overexpressing cells were only used to visualize cilia in live mode, in which the cilium could not otherwise be visualized together with the measurements of the mitochondria membrane potential. Because we did not find differences in the mitochondrial bioenergetics in the mitochondrial subset close to the cilium, we did not consider to include control experiments to see if the expression of ARL13b can affect mitochondrial bioenergetics. In the case of differences, we would have proceeded with further experiments to dissect out the reasons using also other constructs. Importantly, we did not use these constructs for any other further ciliary experiments but only in the experiments to contemporarily visualize in live mode mitochondria and cilia. We also want to clarify that in supplementary figure 3 we stained the fixed cells with an antibody against ARL13b.

The figures would benefit from labels including the stains/antibodies/constructs presented. It is confusing because the organelle markers are different colors in panels of the same figure. Can they be pseudo-colored the same for consistency? We understand the confusion and we labelled the images throughout the figures. We prefer to maintain the original colors because, in our eyes, they are more intuitive (red = RFP, TMRE, green = Venus, GFP).

Finding 2 (Figures 1F-J): Blocking complex I (CI) or ATPase decrease energy levels and ciliation in IMCD3

1. The authors report fewer ciliated cells upon treatment of IMCD3 cells with Rotenone and Oligomycin. However, how the ciliated cells were identified/scored is not reported in the main text, requiring the reader to go to the legends which are brief. We counted the ciliated IMCD3 by visual counting since a software recognition might also recognize cytokinetic bridges and other acetylated tubulin structures in these fast replicating cells. We analyzed visually one field for sample as n=1, which includes approximately 50-100 cells, and we had 3-5 samples (technical replicates) per experiment. So for each experiment n=3-5. Three individual experiments (unless otherwise stated) of this type were performed which heads up to approximately n=15. For each experiment, the values were normalized to the control of the specific experiment since different passage number slightly varied the speed of growth and ciliation. We added all these information in the method section.

“In RPE1, cilia length was measured using the CC bounding box tool in Imaris, which identifies the maximum length for the long axis of the object. Cilia number was quantified by software recognition of number of objects in RPE1 cells or by visual counting in IMCD3 due to the presence of cytokinetic bridges and other acetylated tubulin structures in these cells. For each technical replicate, between 50-100 cells were scored for cilia number and/or length.”

"For ciliary length measurements, every experiment, which included 3-5 technical replicates, was normalized to the DMSO control of that experiment and the length expressed as fold change compared to control. Each experimental set was replicated at least three times unless otherwise stated."

Was an axoneme and basal body marker used to identify cilia? During the final stages of cytokinesis a cytokinetic bridge is present between the daughter cells and may be mistaken as cilia which is part of the reason why a basal body marker is important. In fact, cytokinetic bridges are apparent in some of the images presented. Absolutely correct. In fact we had to visually discriminate the dividing cells with visible cytokinetic bridges from the cilia (for this reason we did not use software recognition as explained above). Moreover, we tested the double staining with ARL13b and acetylated tubulin in IMCD3 but realized that some cilia were negative for ARL13b and, thus, this approach was not feasible. On purpose, we included the images in Fig 1D to show that we did not score the cytokinetic bridges (absence of the white arrow in the previous version of the manuscript). This issue was not present in RPE1 since they did not grow after starvation. To show the distinction between cilia and cytokinetic bridges we labelled the latter with yellow arrows in Fig 1D.

2. It appears the authors used acetylated tubulin antibodies to identify cilia in IMCD3 cells. Why was this approach used as the cells are seemingly already expressing ARL13b that can be used to identify cilia? As the referee previously commented, the expression of ARL13b might change some cell features and thus we used wild-type cells throughout the experiments beside the one on the evaluation of mitochondrial membrane potential and spatial relationship between cilia and mitochondria. Therefore, in all experiments beside the ones in Fig. 1A-C the cells are not expressing ARL13b Venus or RFP in the cilia and thus we used acetylated tubulin for immunostaining and cilia identification.

3. The authors claim a reduced "level" of IFT88 but it is unclear how this was determined i.e. protein expression by Western blot, immunofluorescence etc, and whether the reduced IFT88 is specifically at cilia or at the whole cell level. Moreover, was the level of IFT88 measured relative to ARL13b or ac-tubulin? Thanks for this comment as well. We clarified this point by adding protein expression to the figure. The IFT88 protein expression was measured by WB and the levels were normalized to gamma tubulin since, in our experience, it works as a stable loading control protein upon different cell treatments. Thus, the protein levels of IFT88 are decreased at cellular level and not only at ciliary level corroborating the data on the decrease in cilia number in IMCD3 calculated by counting cilia. Moreover, addressing this reviewer point we realized we forgot to add the WB in the method section, which we added now.

4. Why are "% ciliated cells" presented as a fold change rather than the absolute numbers? As explained above (finding 2 point 1), for each experiment the values were normalized to the control of the specific experiment since at different passages the speed of growth and ciliation varied slightly. Therefore, we expressed the control as 1 (plus the standard deviation of that experiment) even though the real value was roughly between 50 and 80% of ciliated cells depending on the experiment.

This reviewer is surprised by the apparently high statistical difference/low p-values for all of the graphs in Fig 1, particularly given the broad spread of the data points. How did the authors define "n"? How many cells were scored, and how many biological and technical replicates were undertaken? Referring to the high spread of Fig 1G the n was defined as technical plus biological

replicates (as explained just above). Also, in the other experiments in cell lines the n was defined in the same way. Briefly, we scored one field per well and around 5 wells per experiment (technical replicates). Only for titration experiments, we replicated the experiment twice (biological replicates or experimental repetition) while for the other experiments with chemicals the biological replicates were 3-4. This is now briefly described in the statistics section and the n number is described for each experiment in the legends. For animal experiments, each single animal was considered as n=1.

What statistical tests were done on the data? In Fig 1G the statistical analysis used was Mann-Whitney due to non-normal distribution of the data. If using the T-Test the statistical significance would have been even stronger but wouldn't have reflected the data distribution. In the method section, we described that the statistical method was decided based on the calculation of data distribution of each specific sample set.

5. The authors conclude that Rotenone and to a lesser extent Oligomycin impair ciliogenesis. However, from the methods it appears the classical ciliogenesis assay was not undertaken and instead a modified version of a cilia disassembly assay was done. Why do authors conclude the effect of Rotenone and Oligomycin is on ciliogenesis rather than promoting cilia disassembly? Have the authors attempted to discriminate between the two processes? This is another very valid point raised by the referee. At the present, we do not know nor did we investigate whether the reabsorption of the cilia is faster after drug treatments. However, considering that after division the IMCD3 cells rebuild the cilium, we think that the elongation is affected rather than absorption. It is still possible that a faster reabsorption due to ATP deprivation/or oxidative stress is present. To be correct in our conclusions we describe that the treatments affect cilia morphology and not ciliogenesis.

Finding 3 (Figure 2): Reactive oxygen species (ROS) are partly responsible for decreased number of ciliated IMCD3 cells - Not supported We respectfully disagree on the statement that this conclusion is not supported by the data. As explained in detail in the point-by-point response below, we think that using 2 different chemicals which block different sites of the mitochondrial ETC resulting in radicals production (Complex I and ATPase) and the partial rescue of decrease in ciliation with antioxidants in IMCD3 is sufficient to prove the effect of ROS on ciliation.

Have the authors considered that Rotenone has several intracellular actions including the depolymerization of tubulin and that its other actions may contribute to the regulation of cilia? Yes, for this reason we added oligomycin in our experiments in IMCD3 and we performed experiments in glucose or galactose media in RPE1. In IMCD3 the rotenone effect is similar to oligomycin, which does not have the same off-target effect as rotenone, while in RPE1 we could replicate the same results from glucose media (Fig 4E,F) using galactose media (Fig 4J) but with 10 fold less rotenone concentration. The same concentration does not minimally affect cilia in RPE1 in glucose media (Fig 4B). This means that the effect of rotenone on mitochondria is responsible for the effect in RPE, while a similar effect is achieved by rotenone and oligomycin in IMCD3. Moreover, the antioxidant rescue experiments in IMCD3 clearly show that ROS are partially responsible for the loss of cilia, since quenching them increased the ciliated cells compared to rotenone alone.

How specific are the functions of the rescue agents such as Tiron? Tiron is a synthetic analogue of vitamin E, which scavenges superoxide anion. Tiron has been described to be even specific for mitochondria even though the data regarding this are not clear. Tiron has been widely used in the

literature in vivo with beneficial effects in scavenging ROS (e.g. (4)). Beside the use of Tiron, we used other antioxidants and combination of those to maximize the possible beneficial effect and prove that the treatment can partially rescue the cilia phenotype by decreasing ROS.

How was the cell scoring and statistics done? The method is the same as described above. In this experiment we had >3 technical replicates per condition and 4 "biological" replicates. The number of ciliated cells was visually counted. The statistical analysis used was ordinary one-way ANOVA (data are normally distributed) plus Tukey's post hoc test for multiple comparison between each group.

Finding (Figure 3): Ciliation is impaired in the kidney of a diabetic nephropathy mouse model. The authors use db/db mice which display a similar phenotype to diabetic nephropathy in humans, and report impaired ciliation in the kidney. However, apart from noting that diabetic nephropathy kidneys "are subjected to increased oxidative stress and mitochondrial dysfunction" there is no evidence provided to demonstrate a causative association between oxidative stress/mitochondrial dysfunction and the cilia defects. We understand the point raised by the reviewer on the causative effect of oxidative stress and mitochondria dysfunction on the cilia, which would require extensive time course experiments with mice to measure contemporary mitochondria/oxidative phenotype and cilia, and which will be difficult to pinpoint in a specific cell type in vivo. We want to reinforce that we opted for this animal model to translate our findings on IMCD3 into an in vivo model. We used db/db since it is a well-described model regarding kidney nephropathy and in which the presence of oxidative stress during the disease development has been shown before as referenced in the manuscript.

Even though we did not aim to find a causative effect, we understand that more experiments were needed to add information on the mitochondria/oxidative stress at the time point in which cilia were evaluated.

We found a clear increase in oxidative stress gene markers at 24 weeks indicating that the kidneys of this animal model suffer oxidative stress as previously reported (see references in the manuscript). At 8 weeks of age, we found an increase in two antioxidant genes, which could indicate the beginning of a cellular response to oxidative stress. Measuring complex activities in isolated membranes from the same frozen samples in which we measured cilia length, we found a tendency to increased oxygen consumption from complexes I+III+IV and II+III+IV at 8 weeks. However, at 24 weeks of age the complex activity is fairly similar. Although this would not suggest a mitochondrial defect, it seems that mitochondrial complexes activity worsen at 24 weeks of age compared to 8 weeks in db/db mice. The possible increase of mitochondrial activity at 8 weeks of age, which could trigger increased mitochondrial ROS production, could be due to the hyperfiltration shown in these animals during disease development (2) or due to the amount of substrates (hyperglycemia, hyperlipidemia) which could trigger increase in ROS due to excess of energy (see pathogenesis of diabetic complication (3)).

With these experiments, we can conclude that at least oxidative stress was present at 24 weeks when cilia morphology was impaired, while these alterations were not present at 8 weeks where the cilia morphology was similar to the controls. Thus, we correlated at least ROS production with impaired ciliation in vivo in the db/db kidney; while it is not clear how defective the mitochondria are and how much energy production is impaired, although we found a deterioration in some mitochondrial parameters at 24 weeks compared to db/db 8 weeks old.

After the reviewer comments and the experiments performed we completely revised the related section in the manuscript.

Even though we cannot claim any causative effect of oxidative stress/mitochondrial dysfunction causing loss of cilia in this mouse, our in vitro experiment and the observation that oxidative stress is present when cilia morphology is impaired suggests that the in vitro mechanism we found in IMCD3 might be present in vivo. We clarified these concepts in the manuscript. Importantly, oxidative stress has been shown before to be able to elongate the cilia in other work, while in this manuscript we can show for the first time that this might happen in cell-dependent manner also in vivo.

Furthermore, cilia defects in various tissues of db/db mice have previously been reported. To the best of our knowledge, we are not aware of any cilia defects described in the kidney of db/db mice. However, alteration in cilia has been found in hypothalamus and in other animal models of obesity possibly driven by hyperleptinemia (5). This is not the case in our model since hyperleptinemia is present much earlier than cilia loss as described in the manuscript. We want to point out that ciliary control of energy metabolism has been previously suggested but the influence of cell metabolism and especially oxidative stress on cilia is a partial novelty of our work.

Perhaps the authors could consider doing "rescue" studies using primary db/db cells to investigate the role of mitochondria and ROS in the regulation of cilia. Unfortunately, rescue experiments in specific cells in the kidney would be difficult to interpret due to the tissue architecture of the nephron and possible cell dedifferentiation and phenotype change after isolation. For these reasons, we first considered rescue experiments in the animal model with antioxidants. However, since all these approaches have been described to ameliorate the overall mouse phenotype and not specifically the kidney disease, specific conclusions on the ciliation of the kidney would be difficult to interpret. Therefore, we did not pursue this approach.

Finding (Figure 4): Mitochondrial blockade elongates cilia in RPE cells independently of cellular energy status/ROS are responsible for ciliary morphology alteration in RPE cells
The authors report rotenone treatment (but not oligomycin) of RPE cells increased cilia length. Previous studies have reported the similar observations in RPE cells as well as other cell types such as neurons. To the best of our knowledge, we have encountered one publication using enteric neurons from a genetically modified mouse model describing a change on cilia size (decrease in minimum and in some experiments increase of mean or max length) after rotenone treatment (6) plus the paper we cited in the manuscript (7) where the authors studied fibroblasts treated with rotenone. Whereas the first publication studied a genetic mouse model of Alzheimer disease and did not investigate the mechanisms of ciliary alterations, the second work suggests that elongation by rotenone is due to the decrease ATP levels and decrease ciliary reabsorption. Regarding this, we found that the decrease in ATP/ADP ratio was not the reason for the increase in cilia length in RPE1 since oligomycin decrease the ATP/ADP ratio but did not result in longer cilia.

Further, we did not include in the results/discussion, a publication from Bae et al. (8) mainly because of the exponential cell growth they used when performing their experiments in RPE1. Moreover, RPE1 cells have been only used in the first set of experiments where they described an increase in cilia length after blocking complex I or using the uncoupler CCCP. At the same time, they found depolarized mitochondria. Interestingly, in preliminary experiments, which we did not include in the manuscript and confidentially show in the figure below for the reviewer, we found that FCCP (7 μ M) treatment did not change neither cellular ATP in RPE1 nor ciliation. No other experiments have been performed in RPE1 by Bae et al. and in our view the phenotype they found might be due to a cell cycle arrest and build-up of the cilium since in control cells only 20% of the cells were ciliated whereas we found that almost all serum-starved RPE1 cells are ciliated.

The authors report that confluent SH-SY5Y cells did not change the cilia length after treatment with rotenone. However, radical production, antioxidant scavenging as well as mitochondria respiration are dependent on the confluence of cells. We could not reproduce the decrease in ROS in RPE1 with antioxidant treatments when cells were confluent as they did with NAC in the growing SH-SY5Y, which is also in line with the fact that we did not find cytosolic radicals using DCFDA. This suggests that the mitochondrial radicals in the conditions we used are quenched outside the mitochondria. Moreover, it might be that substrate utilization for energy production is very different between confluent (when they also describe elongated network more similar to a live condition) or growing cells, and therefore the mitochondrial blockade of ATP production might trigger different results. In fact, in the SH-SY5Y the authors correlate the mitochondrial blockade to autophagy and apoptosis, which we did not find in our experiments.

We think that the finding from Bae et al. that a single dose of the mitochondrial toxin MPTP which increases also radicals (9) is biologically important and add further value in the context of our work. We therefore added this in the text:

“An increase in cilia length has also been shown after a single dose of mitochondrial inhibitor MPTP in dopaminergic neurons in mice (Bae et al 2019).”

“Interestingly, a single injection of the mitochondrial toxin MPTP in mice, which inhibits complex I and induces ROS, has been show to produce elongation of cilia in dopamine neurons (Bae et al 2019).”

Therefore, it is not clear what the new findings are here. We investigated the effects of mitochondrial dysfunction on cell ciliation in 2 different cell lines with opposite outcomes. We think this is new and biologically important since some cells can cope with decrease mitochondrial function or increase in oxidative stress while other cannot (including during diseases such as aging or diabetes). Until now, the works we mentioned above found an increase in cilia length, which seems to be a mechanism in response to decrease ATP levels or oxidative stress that is not common to all

cell types as we demonstrated here. We think this is important and novel in this context. We show also a complete picture of bioenergetics, including cellular ATP/ADP ratio, mitochondrial and cellular ROS together with cell cycle and cilia morphology and, in RPE1, we induced the increase in cilia length by altering substrate utilization reducing several fold the rotenone concentration. Lastly, we add further novelty and connection between cilia and disease studying cilia in a mouse model of diabetes providing further connection between diabetes, a disease characterized also by mitochondrial dysfunction and oxidative stress, and ciliary dysfunction.

We hope that this explanation can clarify the new findings in this manuscript, which we think are relevant for the field and foster future research. The above-mentioned recent publications show how the field is growing and the need of more data to understand cilia homeostasis in the context of different disease.

Furthermore, I am surprised at the level of statistical significance for the graphs in Figure 4 and would like clearer descriptions of the analysis. We think the reviewer is referring to Fig4 E,F,G,I,J for example. For that reasons we attached the statistical analysis of Fig. 4I as an example. We want to point out that the figure shows mean + confidence interval that show a much larger variation than the standard deviation or standard error of the mean but is for us more intuitive to represent the spread of the data.

Raw data from all technical and biological replicates

C	Rot 800	Oligo 400
1,00	1,42	0,99
1,18	0,99	1,01
0,82	0,78	1,46
0,89	1,98	1,25
1,11	2,18	1,11
0,89	1,98	1,25
1,11	2,18	1,11
1,00	1,45	0,92
0,97	1,86	3,28
1,08	1,97	1,33
0,92	1,91	1,47
1,03	1,93	3,27

Outliers identification with Graphpad prism Grubb's test

C	Rot 800	Oligo 400
1,00	1,42	0,99
1,18	0,99	1,01
0,82	0,78	1,46
0,89	1,98	1,25
1,11	2,18	1,11
0,89	1,98	1,25
1,11	2,18	1,11

1,00	1,45	0,92
0,97	1,86	3,28
1,08	1,97	1,33
0,92	1,91	1,47
1,03	1,93	3,27

C	Rot 800	Oligo 400
1,00	1,42	0,99
1,18	0,99	1,01
0,82	0,78	1,46
0,89	1,98	1,25
1,11	2,18	1,11
0,89	1,98	1,25
1,11	2,18	1,11
1,00	1,45	0,92
0,97	1,86	
1,08	1,97	1,33
0,92	1,91	1,47
1,03	1,93	

Normality Test D'Agostino & Pearson

D'Agostino & Pearson test

K2	0,5873	3,308	1,097
P value	0,7455	0,1913	0,5777
Passed normality test (alpha=0.05)?	Yes	Yes	Yes
P value summary	ns	ns	ns

One-way Ordinary ANOVA

ANOVA summary

F	18,38
P value	<0,0001
P value summary	****
Significant diff. among means (P < 0.05)?	Yes
R squared	0,5425

Brown-Forsythe test

F (DFn, DFd)	3,054 (2, 31)
P value	0,0616
P value summary	ns
Are SDs significantly different (P < 0.05)?	No

Bartlett's test

Bartlett's statistic (corrected) 19,87
P value <0,0001
P value summary ****
Are SDs significantly different (P < 0.05)? Yes

ANOVA table	SS	DF	MS	F (DFn, DFd) F (2, 31) =	P value
Treatment (between columns)	3,312	2	1,656	18,38	P<0,0001
Residual (within columns)	2,793	31	0,09009		
Total	6,105	33			

Data summary
Number of treatments (columns) 3
Number of values (total) 34

Multiple comparison Tukey

Number of families 1
Number of comparisons per family 3
Alpha 0,05

Tukey's multiple comparisons test	Mean Diff,	95,00% CI of diff,	Significant?	Summary	Adjusted P Value
C vs. Rot 800	-0,7197	0,4181 -1,021 to -0,5062	Yes	****	<0,0001 A-B
C vs. Oligo 400	-0,1899	0,1264 0,2135 to	No	ns	0,3152 A-C
Rot 800 vs. Oligo 400	0,5298	0,8461	Yes	***	0,0007 B-C

Test details	Mean 1	Mean 2	Mean Diff,	SE of diff,	n1	n2
C vs. Rot 800	1	1,72	-0,7197	0,1225	12	12
C vs. Oligo 400	1	1,19	-0,1899	0,1285	12	10
Rot 800 vs. Oligo 400	1,72	1,19	0,5298	0,1285	12	10

1. Doke, T., and Susztak, K. (2022) The multifaceted role of kidney tubule mitochondrial dysfunction in kidney disease development. *Trends Cell Biol*
2. Gartner, K. (1978) Glomerular hyperfiltration during the onset of diabetes mellitus in two strains of diabetic mice (c57bl/6j db/db and c57bl/ksj db/db). *Diabetologia* **15**, 59-63
3. Brownlee, M. (2005) The pathobiology of diabetic complications: a unifying mechanism. *Diabetes* **54**, 1615-1625
4. Morgan, A., Galal, M. K., Ogaly, H. A., Ibrahim, M. A., Abd-Elsalam, R. M., and Noshay, P. (2017) Tiron ameliorates oxidative stress and inflammation in titanium dioxide nanoparticles induced nephrotoxicity of male rats. *Biomed Pharmacother* **93**, 779-787
5. Song, D. K., Choi, J. H., and Kim, M. S. (2018) Primary Cilia as a Signaling Platform for Control of Energy Metabolism. *Diabetes Metab J* **42**, 117-127

6. Nguyen, V. T. T., Brucker, L., Volz, A. K., Baumgartner, J. C., Dos Santos Guilherme, M., Valeri, F., May-Simera, H., and Endres, K. (2021) Primary Cilia Structure Is Prolonged in Enteric Neurons of 5xFAD Alzheimer's Disease Model Mice. *Int J Mol Sci* **22**
7. Burkhalter, M. D., Sridhar, A., Sampaio, P., Jacinto, R., Burczyk, M. S., Donow, C., Angenendt, M., Competence Network for Congenital Heart Defects, I., Hempel, M., Walther, P., Pennekamp, P., Omran, H., Lopes, S. S., Ware, S. M., and Philipp, M. (2019) Imbalanced mitochondrial function provokes heterotaxy via aberrant ciliogenesis. *J Clin Invest* **129**, 2841-2855
8. Bae, J. E., Kang, G. M., Min, S. H., Jo, D. S., Jung, Y. K., Kim, K., Kim, M. S., and Cho, D. H. (2019) Primary cilia mediate mitochondrial stress responses to promote dopamine neuron survival in a Parkinson's disease model. *Cell Death Dis* **10**, 952
9. Przedborski, S., Tieu, K., Perier, C., and Vila, M. (2004) MPTP as a mitochondrial neurotoxic model of Parkinson's disease. *J Bioenerg Biomembr* **36**, 375-379

August 29, 2022

RE: Life Science Alliance Manuscript #LSA-2022-01505-TR

Dr. Noah Moruzzi
Karolinska Institute
Molecular medicine and surgery
Anna Steckséns Gata 53
Solna 17176
Sweden

Dear Dr. Moruzzi,

Thank you for submitting your revised manuscript entitled "MITOCHONDRIAL IMPAIRMENT AND INTRACELLULAR REACTIVE OXYGEN SPECIES ALTER PRIMARY CILIA MORPHOLOGY". We would be happy to publish your paper in Life Science Alliance pending final revisions necessary to meet our formatting guidelines.

- It appears the authors are including biological as well as technical replicates in the number of "n", which increased n to 12-15 rather than only including the biological replicates as n ie. n=3. Please include only biological replicates in the number of "n" and modify the statistics accordingly
- please address the remaining Reviewer 1 points
- please make sure the author order in your manuscript and our system match (New author, Ms. Sara Bulgaro missing from the authors list within the ms)
- please add a Category for your manuscript in our system
- please add a callout for each figure to your main manuscript text, some seem to be missing (Fig S4 A, B, C, D, etc)

A. FINAL FILES:

B. MANUSCRIPT ORGANIZATION AND FORMATTING:

Sincerely,

Reviewer #1 (Comments to the Authors (Required)):

In their resubmission, Moruzzi et al. have addressed all previous concerns. A few additional suggestions remain.

1. P.7. Define DCFDA acronym
2. Fig 3H. Separate data of 8 weeks from 24 weeks, since not possible to make out all the conditions in current graph.
3. Fig 1D, 3D, 3J, 4A typo - should be 'acetylated'
4. Fig 4E typo - should be 'length'

September 5, 2022

RE: Life Science Alliance Manuscript #LSA-2022-01505-TRR

Dr. Noah Moruzzi
Karolinska Institute
Molecular medicine and surgery
Anna Steckséns Gata 53
Solna 17176
Sweden

Dear Dr. Moruzzi,

Thank you for submitting your Research Article entitled "MITOCHONDRIAL IMPAIRMENT AND INTRACELLULAR REACTIVE OXYGEN SPECIES ALTER PRIMARY CILIA MORPHOLOGY". It is a pleasure to let you know that your manuscript is now accepted for publication in Life Science Alliance. Congratulations on this interesting work.

DISTRIBUTION OF MATERIALS:

Again, congratulations on a very nice paper. I hope you found the review process to be constructive and are pleased with how the manuscript was handled editorially. We look forward to future exciting submissions from your lab.

Sincerely,
